# META-TARGET DPO: LEARNING ADAPTIVE CONFIDENCE TARGETS VIA META-LEARNING

## ABSTRACT

Direct Preference Optimization (DPO) offers an effective paradigm for aligning Large Language Models (LLMs), yet its performance can be compromised by noisy or ambiguous preference data common in real-world scenarios. Standard DPO formulations often lack mechanisms to adapt to varying levels of reliability across training instances. This paper introduces Meta-Target DPO (MT-DPO), a novel framework that achieves robust preference alignment by dynamically learning adaptive confidence targets for each preference pair. MT-DPO employs a meta-learning approach where an auxiliary confidence module predicts a sample-specific target probability, representing the degree of belief in the observed preference. This module is informed by intrinsic signals, notably perplexity differentials derived from an anchored reference model, indicative of label consistency. Guided by a small, trusted meta-dataset, the confidence module is trained to generate targets that optimally steer the main policy optimization. MT-DPO optimizes the LLM policy using a cross-entropy objective, effectively minimizing the divergence between the policy's implied preference probability and the dynamically learned confidence target for each pair. This allows the learning process to naturally down-weight uncertain instances and potentially rectify contributions from mislabeled data by adapting the target across the full confidence spectrum. Comprehensive experiments on standard alignment benchmarks demonstrate that MT-DPO significantly outperforms vanilla DPO and other robust alignment strategies on both clean and synthetically noisy datasets, showcasing its superior adaptability and effectiveness in handling preference uncertainty through learned target modulation.

## 1 INTRODUCTION

The alignment of Large Language Models (LLMs) with human preferences and societal values is a critical prerequisite for their responsible and beneficial application in diverse domains Bai et al. (2022a); Ouyang et al. (2022); Stiennon et al. (2020); Askell et al. (2021). While Reinforcement Learning from Human Feedback (RLHF) Christiano et al. (2017); Ziegler et al. (2019) has been a cornerstone in this endeavor, its operational complexity, training instability Chung et al. (2024); Eisenstein et al. (2023), and sample inefficiency have spurred research into more direct and data-efficient offline alignment techniques Rafailov et al. (2023); Azar et al. (2023); Ethayarajh et al. (2024).

Direct Preference Optimization (DPO) Rafailov et al. (2023) has gained significant traction as a prominent offline method. It reframes the alignment problem by establishing a direct link between the LLM policy and pairwise preference data Eisenstein et al. (2023), thereby obviating the need for explicit reward model training, a core component of traditional RLHF pipelines Ouyang et al. (2022). DPO optimizes the policy to maximize the likelihood of observed preferences under the Bradley-Terry model Bradley & Terry (1952); Hunter (2004). However, a fundamental limitation of the standard DPO formulation is its implicit assumption of clean, deterministic preference labels, where one response is unequivocally superior to another. This ideal often contrasts sharply with the reality of human preference data, which is frequently characterized by inherent ambiguities, varying degrees of rater consensus, or outright labeling errors due to human biases, task misunderstandings, or annotation inconsistencies Lee et al. (2023); Gao et al. (2024); Chen et al. (2024); Casper et al. (2023). Such imperfections in

the training data can severely undermine the robustness of DPO Mitchell (2023), potentially leading the LLM to learn skewed or incorrect preference distributions.

Efforts to address these challenges have explored various avenues. Some approaches attempt to incorporate soft or probabilistic interpretations of preferences. For instance, Geometric-averaged DPO (GDPO) Furuta et al. (2024) introduced a mechanism to scale the DPO learning signal based on a pre-assigned confidence score, but its original formulation primarily considered scenarios where one response was at least as good as the other, thus not fully addressing the spectrum of label noise. Other methods focus on robust learning from noisy labels, either by adjusting the loss function based on estimated global noise rates Mitchell (2023); Wu et al. (2024) or by developing noise-tolerant objectives such as Robust DPO Liang et al. (2024). Data-centric perspectives attempt to identify and correct noisy instances Song et al. (2022); Han et al. (2020), with methods like PerpCorrect Kong et al. (2024) demonstrating the utility of perplexity differences (PPLDiff) as a signal for detecting inconsistencies in preference labels using an auxiliary model aligned on a small, clean dataset. While these methods offer valuable improvements, they often rely on static heuristics, global noise assumptions, or multi-stage pre-processing steps that may not adapt optimally to the fine-grained, instance-specific variations in label reliability inherent in real-world preference collections Wu et al. (2022).

In this work, we propose Meta-Target DPO (MT-DPO), a novel framework that achieves robust preference alignment by learning to dynamically generate adaptive confidence targets for each preference pair. Instead of relying on fixed or heuristically-derived confidence scores, MT-DPO employs a meta-learning strategy Shu et al. (2019); Jiang et al. (2018); Vettoruzzo et al. (2024) to train an auxiliary confidence module. This module predicts a sample-specific target probability $\hat{p}_n$, representing the model's dynamically inferred belief in the correctness or strength of the observed preference $(y_1 \succ y_2)$. Crucially, this confidence module is informed by intrinsic signals indicative of label quality, with a particular focus on PPLDiff computed using an anchored reference model Kong et al. (2024). The module's parameters are optimized via a meta-objective defined on a small, trusted set of high-quality preference examples $(\mathcal{D}_{meta})$. The LLM policy in MT-DPO is then optimized using a cross-entropy objective, aiming to align its implied preference probability (derived from the DPO log-ratio) with this dynamically learned target $\hat{p}_n$. This adaptive target mechanism allows MT-DPO to act akin to standard DPO for high-confidence, correct pairs (where learned $\hat{p}_n \approx 1$), naturally down-weight ambiguous instances ($\hat{p}_n \approx 0.5$), and effectively rectify the learning signal for likely mislabeled pairs by learning targets approaching zero, thereby navigating the full spectrum of preference confidence and label quality.

Our main contributions are summarized as follows:

- We introduce MT-DPO, a novel meta-learning framework for robust preference optimization that dynamically learns adaptive confidence targets for individual preference pairs.

- We demonstrate how a cross-entropy objective, when coupled with meta-learned targets guided by perplexity differentials, can effectively handle both clean and noisy preference data by adapting the optimization landscape at a sample level.

- Through comprehensive experiments on standard alignment benchmarks, we show that MT-DPO significantly outperforms vanilla DPO and other state-of-the-art robust alignment strategies across diverse data conditions, highlighting its superior adaptability and effectiveness.

## 2 PRELIMINARIES

OThis section provides a concise overview of the foundational concepts underpinning our proposed method, primarily focusing on Direct Preference Optimization (DPO), the formulation of a cross-entropy objective for preference learning, and the notion of perplexity.

### 2.1 DIRECT PREFERENCE OPTIMIZATION (DPO)

Direct Preference Optimization (DPO) Rafailov et al. (2023) has emerged as an influential offline method for aligning Large Language Models (LLMs) with human preferences, circumventing the

need for explicit reward model training. Given a dataset $\mathcal{D} = \{(x^{(i)}, y_w^{(i)}, y_l^{(i)})\}_{i=1}^N$ of preference tuples, where $x$ is a prompt, $y_w$ is the preferred (winner) response, and $y_l$ is the dispreferred (loser) response, DPO directly optimizes the LLM policy $\pi_\theta$. The core idea of DPO is rooted in the Bradley-Terry preference model Bradley & Terry (1952), which posits that the probability of $y_w$ being preferred over $y_l$ can be expressed as $p(y_w \succ y_l|x) = \sigma(r^*(x, y_w) - r^*(x, y_l))$, where $r^*$ is an underlying true reward function and $\sigma(\cdot)$ is the sigmoid function.

DPO establishes a relationship between this implicit reward and the optimal policy $\pi^*$ (derived from standard RLHF objectives) relative to a reference policy $\pi_{ref}$:

$$r^*(x, y_w) - r^*(x, y_l) = \beta \log \frac{\pi^*(y_w|x)\pi_{ref}(y_l|x)}{\pi_{ref}(y_w|x)\pi^*(y_l|x)}, \tag{1}$$

where $\beta$ is a hyperparameter controlling the deviation from $\pi_{ref}$. By substituting the current policy $\pi_\theta$ for $\pi^*$, DPO defines its loss function as the negative log-likelihood of the observed preferences:

$$\mathcal{L}_{\text{DPO}}(\pi_\theta; \pi_{ref}) = -\mathbb{E}_{(x, y_w, y_l) \sim \mathcal{D}} \left[ \log \sigma \left( \beta h_\theta(x, y_w, y_l) \right) \right], \tag{2}$$

where $h_\theta(x, y_w, y_l)$ is the log-probability ratio of the policy $\pi_\theta$ relative to $\pi_{ref}$ for the winner versus the loser responses:

$$h_\theta(x, y_w, y_l) = \log \frac{\pi_\theta(y_w \mid x)\pi_{ref}(y_l \mid x)}{\pi_{ref}(y_w \mid x)\pi_\theta(y_l \mid x)}. \tag{3}$$

Minimizing this loss encourages $\pi_\theta$ to assign a higher likelihood to $y_w$ and a lower likelihood to $y_l$, relative to $\pi_{ref}$, for each prompt $x$.

## 2.2 Cross-Entropy Objective for Preference Learning

The standard DPO loss (Eq. 2), maximizing the log-likelihood of observed preferences, implicitly assumes a target probability of 1 for the preferred response $y_w$ over $y_l$. This inherent assumption makes it sensitive to noisy or ambiguous preference labels common in real-world data, where the observed preference may not hold with full certainty.

To overcome this limitation and enable robust learning under uncertainty, we reformulate the preference learning objective using a binary cross-entropy (BCE) loss. This formulation explicitly separates the model's predicted preference probability, $P_{\text{model}}(y_1 \succ y_2|x; \theta)$, from the desired target confidence, $\hat{p} = p_{\text{target}}(y_1 \succ y_2|x)$:

$$P_{\text{model}}(y_1 \succ y_2|x; \theta) = \sigma(\beta h_\theta(x, y_1, y_2)). \tag{4}$$

The BCE loss then minimizes the divergence between the model's prediction and the target $\hat{p}$:

$$\mathcal{L}_{\text{BCE}}(\pi_\theta; \pi_{ref}, \hat{p}) = -\mathbb{E}_{(x, y_1, y_2, \hat{p}) \sim \mathcal{D}} \Big[ \hat{p} \log P_{\text{model}}(y_1 \succ y_2|x; \theta) \\ + (1 - \hat{p}) \log(1 - P_{\text{model}}(y_1 \succ y_2|x; \theta)) \Big]. \tag{5}$$

Crucially, this BCE framework provides the necessary flexibility to move beyond the fixed target of 1 implicit in standard DPO. By allowing $\hat{p}$ to vary, we can modulate the learning signal based on the perceived reliability of each preference pair, aiming to align the policy $\pi_\theta$'s implied probability $P_{\text{model}}$ with a more nuanced confidence level.

The challenge, then, lies in determining appropriate values for the target $\hat{p}$. One approach, exemplified by Conservative DPO (cDPO) Mitchell (2023), involves using *static*, pre-defined targets derived from global assumptions like a known noise rate (e.g., setting $\hat{p} = 1 - \epsilon > 0.5$, if $y_1$ is the nominal winner and $\epsilon$ is the assumed noise rate). While this offers some robustness, relying on global, static targets fails to capture the instance-specific variations in label quality.

In contrast, our proposed method, MT-DPO, leverages the flexibility of the BCE objective (Eq. 5) in a fundamentally different way. Instead of pre-specifying $\hat{p}$, MT-DPO views it as a dynamic, sample-specific confidence target $\hat{p}_n$ spanning the full spectrum [0, 1]. Most importantly, MT-DPO *learns* this target $\hat{p}_n$ adaptively for each instance $n$ via a meta-learning process guided by intrinsic signals of data quality. This learned, adaptive nature of the target is central to MT-DPO's ability to robustly handle diverse and potentially non-uniform data imperfections, forming a core contribution of our work.

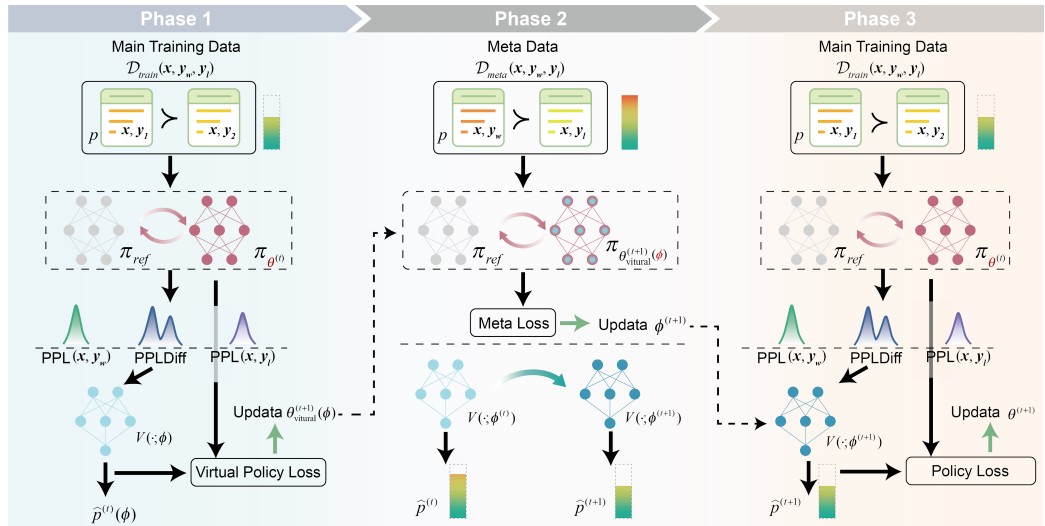

Figure 1: Overall architecture of the Meta-Target DPO (MT-DPO) framework. The confidence prediction module $V(\cdot; \phi)$ generates a sample-specific target $\hat{p}$ based on PPLDiff. The policy LLM $\pi_\theta$ is updated using a cross-entropy loss with this dynamic target. The confidence module itself is updated via a meta-learning loop guided by a clean meta-dataset $\mathcal{D}_{meta}$.

## 3 METHODOLOGY

Building upon the preliminaries, we now introduce our proposed framework, Meta-Target DPO (MT-DPO), for robust preference alignment. MT-DPO leverages a meta-learning approach to dynamically generate adaptive confidence targets for a cross-entropy based DPO objective, guided by intrinsic signals of preference consistency.

### 3.1 OVERALL FRAMEWORK

Aligning Large Language Models (LLMs) with human preferences under noisy or ambiguous data conditions presents a significant challenge, as static rules or global noise estimates often lack adaptability. Our proposed Meta-Target DPO (MT-DPO) framework tackles this by learning an adaptive, sample-specific confidence target, $\hat{p}_n$, for each preference pair. This target dynamically reflects the inferred belief in the correctness of the observed preference.

As depicted in Figure 1, MT-DPO revolves around three interacting components: a base policy LLM ($\pi_\theta$) optimized via a cross-entropy loss with these dynamic targets; a confidence prediction module ($V(\cdot; \phi)$) that generates $\hat{p}_n$ from instance features; and a meta-learning procedure that refines $\phi$ using a small, clean meta-dataset ($\mathcal{D}_{meta}$). This orchestrated interplay ensures that the policy learns from adaptively calibrated preference signals, enhancing alignment robustness. Subsequent sections will detail each component of this framework.

### 3.2 ADAPTIVE CROSS-ENTROPY OBJECTIVE FOR POLICY LEARNING

The policy LLM $\pi_\theta$ in MT-DPO is trained to minimize a cross-entropy loss, similar in form to Eq. 5, but with the crucial difference that the target probability $\hat{p}$ is now a sample-specific, learned value $\hat{p}_n$ predicted by the confidence module $V(I_n; \phi)$, where $I_n$ represents the input features for sample $n$. Let $P_{\text{model}}(y_1^{(n)} \succ y_2^{(n)} | x^{(n)}; \theta)$ be the policy's implied preference probability for sample $n$, as defined in Eq. 4. The objective for updating $\pi_\theta$ is:

$$\mathcal{L}_{\text{policy}}(\theta, \phi) = -\mathbb{E}_{(x^{(n)}, y_1^{(n)}, y_2^{(n)}) \sim \mathcal{D}_{train}} \Big[ \hat{p}_n \log P_{\text{model}}(y_1^{(n)} \succ y_2^{(n)} | x^{(n)}; \theta)$$
$$+ (1 - \hat{p}_n) \log(1 - P_{\text{model}}(y_1^{(n)} \succ y_2^{(n)} | x^{(n)}; \theta)) \Big], \quad (6)$$

where $\hat{p}_n = V(I_n; \phi)$. By dynamically adjusting $\hat{p}_n$, this objective allows MT-DPO to adaptively modulate the learning process. If the confidence module assigns $\hat{p}_n \approx 1$, the loss term encourages $\pi_\theta$ to strongly prefer $y_1^{(n)}$ over $y_2^{(n)}$. Conversely, if $\hat{p}_n \approx 0$, it encourages the opposite preference,

effectively correcting for a perceived label error. Intermediate values of $\hat{p}_n$ lead to a down-weighting of the sample's influence, suitable for ambiguous or highly uncertain pairs.

### 3.3 Confidence Module Input: Perplexity Differentials

The effectiveness of the confidence module $V(\cdot; \phi)$ hinges on informative input features $I_n$ that correlate with preference label quality or consistency. Inspired by the findings in Kong et al. (2024), we primarily utilize the perplexity difference (PPLDiff) between the two responses in a preference pair. Given a preference pair $(x^{(n)}, y_1^{(n)}, y_2^{(n)})$ and an anchored reference language model $\pi_{anchor}$ (which could be the initial SFT model $\pi_{ref}$, or a separate model periodically updated on clean data, as in Kong et al. (2024)), the PPLDiff is computed as:

$$\text{PPLDiff}^{(n)} = \log \text{PPL}(x^{(n)}, y_1^{(n)}; \pi_{anchor}) - \log \text{PPL}(x^{(n)}, y_2^{(n)}; \pi_{anchor}). \tag{7}$$

Here, $\text{PPL}(x, y; \pi_{anchor})$ denotes the perplexity of the concatenated sequence $[x; y]$ under the model $\pi_{anchor}$, calculated as the exponential of the average negative log-likelihood of the sequence. Intuitively, if $y_1^{(n)}$ is genuinely preferred and consistent with aligned language, its PPL should be lower than that of $y_2^{(n)}$, leading to a negative PPLDiff. Conversely, if $y_1^{(n)}$ is noisy (i.e., $y_2^{(n)}$ is actually better), its PPL might be higher, resulting in a positive PPLDiff. A PPLDiff close to zero might indicate ambiguity or similar quality. This PPLDiff value serves as a primary input $I_n$ to our confidence module $V(\cdot; \phi)$, which is typically a small multi-layer perceptron (MLP) outputting a value in $[0, 1]$. Other auxiliary features, such as response lengths or initial policy log-probabilities, could also be incorporated into $I_n$. The choice of $\pi_{anchor}$ is crucial; in our main experiments, we use the SFT model, and we provide a detailed analysis of this choice in Appendix E.1.

### 3.4 Meta-Learning Procedure for Confidence Module

The parameters $\phi$ of the confidence module $V(\cdot; \phi)$ are learned through a meta-optimization process, akin to the strategy in Meta-Weight-Net Shu et al. (2019). This process requires a small, trusted meta-dataset $\mathcal{D}_{meta} = \{(x^{(m)}, y_w^{(m)}, y_l^{(m)})\}_{m=1}^M$ consisting of clean preference pairs, where $y_w^{(m)}$ is definitively preferred over $y_l^{(m)}$.

The meta-learning proceeds iteratively within each training iteration $t$. The process involves three key steps:

First, for a mini-batch $\mathcal{B}_{train}$ from $\mathcal{D}_{train}$, confidence targets $\hat{p}_n^{(t)} = V(I_n; \phi^{(t)})$ are computed using the current confidence module parameters $\phi^{(t)}$. A hypothetical (virtual) update of the policy parameters $\theta^{(t)}$ is then performed by taking one gradient step on the policy loss $\mathcal{L}_{\text{policy}}(\theta^{(t)}, \phi^{(t)})$ with respect to $\theta^{(t)}$ on $\mathcal{B}_{train}$. This results in the virtual policy parameters:

$$\theta_{\text{virtual}}^{(t+1)}(\phi^{(t)}) = \theta^{(t)} - \alpha \nabla_{\theta^{(t)}} \mathcal{L}_{\text{policy}}(\theta^{(t)}, \phi^{(t)}). \tag{8}$$

It is crucial to note that $\theta_{\text{virtual}}^{(t+1)}$ remains a function of $\phi^{(t)}$ because the policy loss $\mathcal{L}_{\text{policy}}$ depends on the targets $\hat{p}_n^{(t)}$, which are generated by $V(I_n; \phi^{(t)})$.

Next, the performance of this virtual policy $\theta_{\text{virtual}}^{(t+1)}(\phi^{(t)})$ is evaluated on a mini-batch $\mathcal{B}_{meta}$ sampled from the clean meta-dataset $\mathcal{D}_{meta}$. The meta-objective is the standard DPO loss, as preferences in $\mathcal{D}_{meta}$ are assumed to be clean (effectively implying $\hat{p} = 1$ for these pairs). This meta-loss, which depends on $\phi^{(t)}$ through $\theta_{\text{virtual}}^{(t+1)}$, is defined as:

$$\mathcal{L}_{\text{meta}}(\phi^{(t)}) = \mathcal{L}_{\text{DPO}}(\pi_{\theta_{\text{virtual}}^{(t+1)}(\phi^{(t)})}; \pi_{ref}). \tag{9}$$

The parameters $\phi$ of the confidence module are then updated by descending the gradient of this meta-loss with respect to $\phi^{(t)}$:

$$\phi^{(t+1)} = \phi^{(t)} - \eta \nabla_{\phi^{(t)}} \mathcal{L}_{\text{meta}}(\phi^{(t)}), \tag{10}$$

where $\eta$ is the meta-learning rate. The gradient $\nabla_{\phi^{(t)}} \mathcal{L}_{\text{meta}}(\phi^{(t)})$ is computed via backpropagation through the virtual policy update step (Eq. 8).

Finally, the actual policy parameters $\theta$ are updated. This update uses the newly refined confidence module parameters $\phi^{(t+1)}$ to compute fresh confidence targets $\hat{p}_n^{(t+1)} = V(I_n; \phi^{(t+1)})$ for the original training mini-batch $\mathcal{B}_{train}$. The policy parameters are then updated by descending the gradient of $\mathcal{L}_{\text{policy}}(\theta^{(t)}, \phi^{(t+1)})$ with respect to $\theta^{(t)}$:

$$\theta^{(t+1)} = \theta^{(t)} - \alpha \nabla_{\theta^{(t)}} \mathcal{L}_{\text{policy}}(\theta^{(t)}, \phi^{(t+1)}). \tag{11}$$

This iterative process enables $V(\cdot; \phi)$ to learn to assign confidence targets $\hat{p}_n$ that, when utilized in $\mathcal{L}_{\text{policy}}$, guide the policy $\pi_\theta$ towards improved performance on trusted, clean preference data. The overall training algorithm is summarized in Appendix A. Further theoretical analysis, including a generalization bound for the meta-learned confidence module, is provided in Appendix B.

## 4 EXPERIMENTS

We conduct experiments to evaluate our proposed Meta-Target DPO (MT-DPO) framework, focusing on its effectiveness and robustness in aligning LLMs with human preferences compared to relevant baselines.

### 4.1 EXPERIMENTAL SETUP

**Datasets and Noise Simulation.** Our evaluations are performed on two standard preference datasets: Golden HH dataset Bai et al. (2022a;b) and OASST1 Stiennon et al. (2020); Völske et al. (2017), using their standard splits. To assess robustness, we introduce symmetric label noise to the training sets by randomly flipping preference labels with probabilities $\epsilon_{\text{noise}} \in \{0\% \text{ (clean)}, 10\%, 20\%, 30\%, 40\%\}$, following protocols from Mitchell (2023); Kong et al. (2024); Liang et al. (2024). For MT-DPO's meta-learning, a small, clean meta-dataset $\mathcal{D}_{meta}$ is sampled and held out, as suggested by Kong et al. (2024); Shu et al. (2019).

**Baselines and Evaluation.** We compare MT-DPO against several strong baselines: Vanilla DPO Rafailov et al. (2023), GDPO (with fixed $\hat{p} = 0.75$) Furuta et al. (2024), Conservative DPO (cDPO, with oracle $\epsilon_{\text{noise}}$ for noisy settings and $\hat{p} = 1.0$ for clean) Mitchell (2023), Robust DPO (rDPO) Liang et al. (2024), and PerpCorrect-DPO Kong et al. (2024) (DPO on data denoised by PerpCorrect using $\mathcal{D}_{meta}$). All DPO-based methods share the same base LLM architecture and an SFT-derived reference policy $\pi_{ref}$ for fair comparison. Performance is primarily measured by win rate (%) against the SFT model on test sets, evaluated by an independent LLM judge (GPT-4) following Rafailov et al. (2023); Lee et al. (2023). Further details on dataset processing, specific baseline configurations (including GDPO variants), and precise evaluation protocols are provided in Appendix C.

### 4.2 IMPLEMENTATION DETAILS

**Base Models and SFT:** We use pre-trained LLMs such as Llama-2-7B Touvron et al. (2023) or phi-2 Javaheripi et al. (2023) as our base models. Prior to DPO-style alignment, these models are first supervised fine-tuned (SFT) on the preferred responses ($y_w$) from the clean training portion of the respective datasets. This SFT model then serves as the initial policy $\theta^{(0)}$ and the reference policy $\pi_{ref}$ for all DPO methods, including MT-DPO. The SFT model also acts as the default anchored model $\pi_{anchor}$ for PPLDiff computation in MT-DPO and PerpCorrect-DPO, unless otherwise specified.

**MT-DPO Configuration and Training:** Our MT-DPO's confidence prediction module $V(\cdot; \phi)$ is implemented as a small MLP (typically with one or two hidden layers and a final sigmoid activation) with the perplexity difference (PPLDiff), potentially normalized, as its primary input $I_n$. The SFT model serves as the default anchored model $\pi_{anchor}$ for PPLDiff computation. We use the AdamW optimizer for both the policy LLM $\theta$ (learning rates typically $1e^{-6}$ to $5e^{-5}$) and the confidence module $\phi$ (learning rates typically $1e^{-4}$ to $1e^{-3}$). The DPO hyperparameter $\beta$ is consistently set to 0.1 across all DPO variants.

All models, including baselines, are trained for a fixed number of epochs or steps on the respective training datasets. We utilize standard deep learning libraries such as PyTorch, along with libraries from Hugging Face Transformers and TRL von Werra et al. (2020), for implementation. All experiments are conducted on NVIDIA A100 GPUs. Comprehensive hyperparameter settings for all methods and datasets, including specific learning rates and $\mathcal{D}_{meta}$ size, are detailed in Appendix D.

Table 1: Win Rates (%) of Llama-2-7B Against SFT on Golden HH and OASST1 Datasets.

| Method | Golden HH | | | | | OASST1 | | | | |
|---|---|---|---|---|---|---|---|---|---|---|
| | Clean (0%) | 10% | 20% | 30% | 40% | Clean (0%) | 10% | 20% | 30% | 40% |
| Vanilla DPO | 97.22 | 92.53 | 82.62 | 68.50 | 53.15 | 97.17 | 96.64 | 92.71 | 90.21 | 86.29 |
| GDPO ($\hat{p} = 0.75$) | 97.57 | 97.15 | 95.53 | 94.26 | 91.21 | 97.52 | 97.06 | 94.21 | 93.08 | 92.73 |
| cDPO (Oracle $\epsilon/\hat{p} = 1$) | 97.38 | 96.04 | 90.85 | 83.23 | 65.60 | 97.67 | 96.18 | 93.63 | 90.62 | 88.02 |
| rDPO | 97.21 | 96.65 | 95.22 | 93.90 | 90.45 | 97.76 | 95.92 | 93.73 | 92.05 | 90.62 |
| PerpCorrect-DPO | 97.87 | 97.51 | 96.24 | 95.53 | 94.92 | 98.05 | 96.38 | 94.04 | 93.99 | 93.17 |
| **MT-DPO (Ours)** | **98.02** | **97.85** | **97.31** | **96.58** | **95.50** | **98.55** | **97.17** | **95.54** | **94.92** | **94.28** |

Table 2: Win Rates (%) of Phi-2 Against SFT on Golden HH and OASST1 Datasets.

| Method | Golden HH | | | | | OASST1 | | | | |
|---|---|---|---|---|---|---|---|---|---|---|
| | Clean (0%) | 10% | 20% | 30% | 40% | Clean (0%) | 10% | 20% | 30% | 40% |
| Vanilla DPO | 96.50 | 93.19 | 85.57 | 73.07 | 54.98 | 69.12 | 66.94 | 62.61 | 58.44 | 52.42 |
| GDPO ($\hat{p} = 0.75$) | 97.07 | 97.54 | 96.08 | 94.52 | 85.39 | 68.73 | 67.93 | 63.58 | 59.88 | 53.05 |
| cDPO (Oracle $\epsilon/\hat{p} = 1$) | 97.56 | 97.21 | 92.63 | 81.05 | 66.72 | 69.30 | 67.30 | 61.44 | 54.87 | 49.21 |
| rDPO | 97.01 | 96.49 | 95.73 | 93.34 | 84.55 | 67.16 | 63.95 | 59.47 | 56.45 | 45.20 |
| PerpCorrect-DPO | 98.18 | 98.17 | 97.05 | 97.66 | 96.39 | 72.55 | 71.34 | 69.04 | 68.27 | 68.49 |
| **MT-DPO (Ours)** | **98.63** | **98.33** | **97.58** | **98.49** | **97.91** | **74.69** | **72.54** | **71.15** | **70.57** | **69.91** |

### 4.2.1 MAIN PERFORMANCE COMPARISON

We now present the empirical results of MT-DPO and compare its performance against the baselines on both clean and noisy preference datasets. For the noisy data conditions ($\epsilon_{\text{noise}} \in \{10\%, 20\%, 30\%, 40\%\}$), results for baseline methods (Vanilla DPO, cDPO, rDPO, PerpCorrect-DPO) are primarily adapted from Kong et al. (2024) Kong et al. (2024) where applicable for the respective models and datasets, unless specified otherwise by our own baseline runs. All results for our proposed MT-DPO and GDPO ($\hat{p} = 0.75$), as well as all results on clean (0% noise) data across all methods, are from our own reproducible experiments. The primary metric is the win rate (%) against the SFT model, evaluated by GPT-4. All results are consolidated in Table 1 for Llama-2-7B and Table 2 for Phi-2. The improvements of our implemented models (MT-DPO and GDPO) over our SFT baseline were confirmed to be statistically significant (Wilcoxon signed-rank test, p ¡ 0.01). Comparisons with other baseline methods rely on their reported aggregate performance metrics and observed differences in win rates.

As shown in Tables 1 and 2, our proposed MT-DPO consistently achieves state-of-the-art (SOTA) performance across all evaluated conditions. On clean datasets (0% noise), MT-DPO demonstrates top-tier results, indicating its adaptive mechanism effectively handles ideal conditions without performance degradation; for example, it achieved the highest clean data win rate of 98.63% with the Phi-2 model on the Golden HH dataset.

The **robustness of MT-DPO under noisy conditions** is particularly compelling. While Vanilla DPO's performance significantly deteriorates with increasing noise, MT-DPO maintains substantially higher win rates. It consistently outperforms other robust baselines, including cDPO (Oracle), rDPO, and PerpCorrect-DPO, across most noise levels, models, and datasets. This underscores the advantage of MT-DPO's dynamic, meta-learned target adaptation strategy in navigating imperfect preference data. For example, under 40% noise with Llama-2-7B on Golden HH, MT-DPO (95.50%) significantly surpasses Vanilla DPO (53.15%) and also leads other robust methods like PerpCorrect-DPO (94.92%).

Notably, GDPO ($\hat{p} = 0.75$) also exhibits strong robustness, often outperforming cDPO and rDPO on the Golden HH dataset, suggesting the efficacy of a well-chosen conservative target. However, MT-DPO generally provides further improvements by adapting these targets at a sample level.

In summary, the empirical results strongly validate MT-DPO as a highly effective and robust method for aligning LLMs, demonstrating superior performance and resilience to noise compared to existing approaches.

### 4.2.2 ABLATION STUDIES

To understand the contribution of different components of MT-DPO, we conducted ablation studies focusing on the impact of the meta-dataset size and the input features for the confidence module. These experiments were performed on the Golden HH dataset with 30% noise using the Llama-2-7B model.

**Impact of Meta-Dataset Size.** We investigated MT-DPO's sensitivity to the size of the clean meta-dataset, $|\mathcal{D}_{meta}|$. Table 3 presents the win rates as $|\mathcal{D}_{meta}|$ varies.

Table 3: Impact of meta-dataset size ($|\mathcal{D}_{meta}|$) on MT-DPO performance (Win rate % vs SFT) on Golden HH with $\epsilon_{\text{noise}} = 30\%$ using Llama-2-7B.

| $|\mathcal{D}_{meta}|$ | 25 | 50 | 100 | 150 | 200 |
|---|---|---|---|---|---|
| MT-DPO Win Rate (%) | 94.66 | 95.37 | 96.29 | **96.58** | 96.70 |

The results indicate that MT-DPO achieves strong performance even with a very small meta-dataset; utilizing only 50 samples for $\mathcal{D}_{meta}$ yielded a win rate of 95.37%. This confirms the data efficiency of the meta-learning guidance, consistent with findings in Shu et al. (2019); Kong et al. (2024). While performance improves with more meta-data, the gains diminish beyond approximately 150 samples (96.58%), suggesting that a modest amount of clean data is sufficient for the confidence module to learn effective target calibration strategies. Increasing $|\mathcal{D}_{meta}|$ to 200 resulted in only a marginal improvement to 96.70%.

**Importance of PPLDiff as Input Feature.** To verify the utility of PPLDiff as the primary input feature $I_n$ for the confidence module, we compared the full MT-DPO against two variants: (1) MT-DPO-NoFeat, where the confidence module learns a global $\hat{p}$ without sample-specific input features, and (2) MT-DPO-Loss, which uses the sample's DPO loss from a prior iteration as input, similar to Shu et al. (2019). Table 4 shows these results.

Table 4: Impact of input features for the confidence module on MT-DPO performance (Win rate % vs SFT) on Golden HH with $\epsilon_{\text{noise}} = 30\%$ using Llama-2-7B.

| Feature for Confidence Module $V(\cdot; \phi)$ | Win Rate (%) |
|---|---|
| No Features (Global $\hat{p}$ learned) | 93.58 |
| DPO Loss Value (MT-DPO-Loss) | 94.83 |
| PPLDiff (Full MT-DPO) | **96.58** |

The results in Table 4 clearly demonstrate that utilizing PPLDiff as an input feature significantly enhances MT-DPO's performance. The full MT-DPO (96.58%) substantially outperforms both the MT-DPO-Loss variant (94.83%) and the MT-DPO-NoFeat variant (93.58%). This underscores the effectiveness of PPLDiff as a reliable indicator of preference consistency and label quality, enabling the confidence module to learn more accurate and beneficial sample-specific targets $\hat{p}_n$.

### 4.2.3 ANALYSIS OF LEARNED CONFIDENCE TARGETS

To gain deeper insight into the adaptive mechanism of MT-DPO, we analyzed the behavior of the learned confidence targets $\hat{p}_n$ generated by the trained confidence module $V(\cdot; \phi)$. This investigation focuses on understanding how the module assigns these targets in response to varying perceived label quality and how these assignments correlate with PPLDiff. For this analysis, we utilized a subset of the Golden HH training data where 30% symmetric label noise was introduced. After training MT-DPO (Llama-2-7B), we passed samples from this noisy training subset through the converged confidence module.

Figure 2 presents a combined visualization. Figure 2a illustrates the distributions of learned $\hat{p}_n$ for samples known to be "Clean" (original label preserved) versus "Noisy (Flipped)" (label intentionally flipped). A clear separation is anticipated: clean samples should have $\hat{p}_n$ concentrated near 1.0, while noisy samples should have $\hat{p}_n$ skewed towards 0.0. Figure 2b presents a scatter plot of $\hat{p}_n$ against PPLDiff. A negative correlation is expected: instances with strongly negative PPLDiff

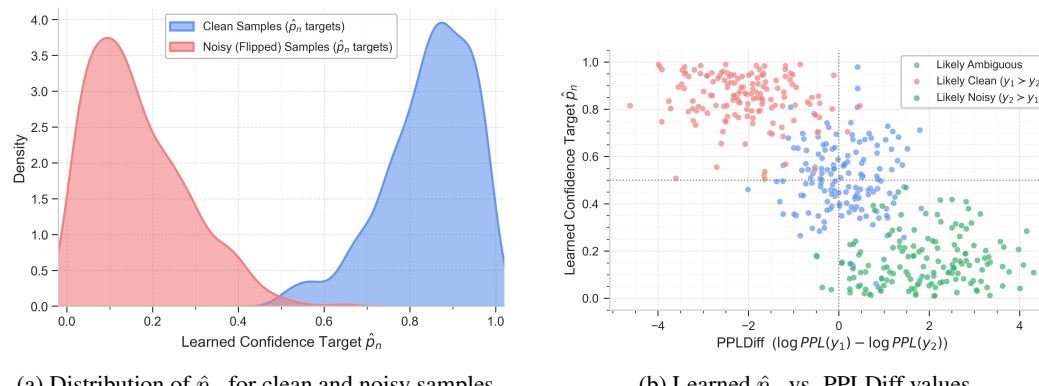

(a) Distribution of $\hat{p}_n$ for clean and noisy samples.

(b) Learned $\hat{p}_n$ vs. PPLDiff values.

Figure 2: Analysis of learned confidence targets $\hat{p}_n$ from MT-DPO on Golden HH with 30% noise (Llama-2-7B). (a) Distinct distributions of $\hat{p}_n$ for known clean and noisy samples. (b) Negative correlation between PPLDiff and learned $\hat{p}_n$.

(suggesting $y_1$ is more likely under $\pi_{anchor}$) should receive $\hat{p}_n \approx 1.0$, while positive PPLDiff values should correlate with $\hat{p}_n \approx 0.0$.

This qualitative analysis is intended to support the hypothesis that the meta-learned confidence module in MT-DPO learns an interpretable and effective mapping from PPLDiff to adaptive confidence targets. This ability to differentiate preference signals and modulate the learning objective accordingly is key to MT-DPO's robust performance. Further details on this analysis and illustrative case studies demonstrating the mechanism in action are provided in Appendix E.2 and F.

## 4.3 CONCLUSION

We introduced Meta-Target DPO (MT-DPO), a novel meta-learning framework for robust preference alignment. MT-DPO dynamically learns adaptive, sample-specific confidence targets for a cross-entropy DPO objective, guided by signals like perplexity differentials and a small clean meta-dataset. This allows intelligent modulation of the learning signal per preference pair, effectively handling diverse label quality. Experiments showed MT-DPO significantly outperforms standard DPO and robust baselines on clean and noisy datasets, highlighting its adaptability in achieving reliable LLM alignment under uncertainty. While MT-DPO demonstrates strong results, its efficacy relies on the quality of the small meta-dataset, and the meta-learning process introduces some computational overhead compared to simpler DPO variants. Future work includes exploring richer input features for the confidence module, investigating sensitivity to meta-dataset characteristics, and extending MT-DPO to other preference learning paradigms. MT-DPO offers a principled direction for building more resilient AI systems from imperfect human feedback.

## ETHICS STATEMENT

In accordance with ICLR policy, we disclose that large language models (LLMs) were employed as writing assistants during the preparation of this paper. Their primary function was to support grammar correction and language refinement, with the goal of improving the overall readability of the manuscript. All core ideas and analyses were conceived and developed solely by the human authors, who assume full responsibility for the final content of the paper.

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

# A ALGORITHM DETAILS

---

**Algorithm 1** MT-DPO Training Procedure

---

1: **Input:** Initial policy $\theta^{(0)}$, initial confidence module $\phi^{(0)}$, reference $\pi_{ref}$, (optional) anchor $\pi_{anchor}$, training data $\mathcal{D}_{train}$, meta-data $\mathcal{D}_{meta}$, learning rates $\alpha, \eta$, iterations $T$.
2: **Output:** Trained policy parameters $\theta^{(T)}$.
3: **for** $t = 0, 1, \ldots, T - 1$ **do**
4:     Sample mini-batch $\mathcal{B}_{train}$ from $\mathcal{D}_{train}$; Sample mini-batch $\mathcal{B}_{meta}$ from $\mathcal{D}_{meta}$.
5:     For $n \in \mathcal{B}_{train}$, compute features $I_n$ (e.g., PPLDiff using $\pi_{anchor}$).
6:     // — Virtual Policy Update —
7:     Compute confidence targets $\hat{p}_n^{(t)} = V(I_n; \phi^{(t)})$ for samples in $\mathcal{B}_{train}$.
8:     Compute virtual policy parameters $\theta_{\text{virtual}}^{(t+1)}(\phi^{(t)})$ using $\mathcal{B}_{train}$ with targets $\hat{p}_n^{(t)}$ and policy loss $\mathcal{L}_{\text{policy}}(\theta^{(t)}, \phi^{(t)})$.
9:     // — Confidence Module Update —
10:     Update confidence module parameters $\phi^{(t+1)}$ by minimizing meta-loss $\mathcal{L}_{\text{meta}}(\phi^{(t)})$.
11:     // — Actual Policy Update —
12:     Compute fresh confidence targets $\hat{p}_n^{(t+1)} = V(I_n; \phi^{(t+1)})$ for samples in $\mathcal{B}_{train}$.
13:     Update policy parameters $\theta^{(t+1)}$ using $\mathcal{B}_{train}$ with targets $\hat{p}_n^{(t+1)}$ and policy loss $\mathcal{L}_{\text{policy}}(\theta^{(t)}, \phi^{(t+1)})$.
14: **end for**
15: **return** $\theta^{(T)}$.

---

# B THEORETICAL ANALYSIS ON THE ALGORITHM

## B.1 WEIGHTING SCHEME INTERPRETATION OF THE CONFIDENCE MODULE UPDATE

The parameters $\phi$ of the confidence module $V(\cdot; \phi)$ in MT-DPO are optimized via meta-learning. This process implicitly learns to assign an optimal target confidence $p_n = V(I_n; \phi)$ for each preference pair $(x, y_w, y_l)$ by adjusting $\phi$ through its gradient updates.

The update of $\phi$ is guided by the performance of a virtual policy $\theta_{\text{virtual}}^{(t+1)}(\phi^{(t)})$ on the meta-dataset $B_{\text{meta}}$. The virtual policy is obtained by a one-step gradient descent of the main policy $\theta^{(t)}$ on the policy loss $L_{\text{policy}}(\theta^{(t)}, \phi^{(t)})$, which itself depends on $p_n^{(t)}$:

$$\theta_{\text{virtual}}^{(t+1)}(\phi^{(t)}) = \theta^{(t)} - \alpha \nabla_{\theta^{(t)}} L_{\text{policy}}(\theta^{(t)}, \phi^{(t)}). \tag{12}$$

The confidence module parameters $\phi$ are then updated as:

$$\phi^{(t+1)} = \phi^{(t)} - \eta \nabla_{\phi^{(t)}} L_{\text{meta}}(\phi^{(t)}), \tag{13}$$

where $L_{\text{meta}}(\phi^{(t)}) = L_{\text{DPO}}(\pi_{\theta_{\text{virtual}}^{(t+1)}(\phi^{(t)})}; \pi_{\text{ref}})$ is evaluated on $B_{\text{meta}}$.

Applying the chain rule to $\nabla_{\phi^{(t)}} L_{\text{meta}}(\phi^{(t)})$ yields:

$$\nabla_{\phi^{(t)}} L_{\text{meta}}(\phi^{(t)}) = \left( \frac{\partial L_{\text{meta}}}{\partial \theta_{\text{virtual}}^{(t+1)}} \right)^T \frac{\partial \theta_{\text{virtual}}^{(t+1)}}{\partial \phi^{(t)}}. \tag{14}$$

Substituting $\frac{\partial \theta_{\text{virtual}}^{(t+1)}}{\partial \phi^{(t)}} = -\alpha \nabla_{\theta^{(t)}, \phi^{(t)}}^2 L_{\text{policy}}(\theta^{(t)}, \phi^{(t)})$, the update direction for $\phi$ is approximately proportional to:

$$\Delta \phi^{(t)} \propto \left( \nabla_{\theta_{\text{virtual}}^{(t+1)}} L_{\text{meta}} \right)^T \left( \nabla_{\theta^{(t)}, \phi^{(t)}}^2 L_{\text{policy}} \right). \tag{15}$$

The term $\left( \nabla_{\theta_{\text{virtual}}^{(t+1)}} L_{\text{meta}} \right)^T \left( \nabla_{\theta^{(t)}, \phi^{(t)}}^2 L_{\text{policy}} \right)$ in Eq. 15 reflects the "alignment" between the change in policy learning direction—induced by adjusting target confidences $p_n$ (controlled by $\phi$)—and the

desired direction for meta-objective optimization. If adjusting $p_n$ steers the main policy's learning direction closer to the optimal direction on $B_{\text{meta}}$, such an adjustment to $\phi$ is reinforced. This mechanism enables $V(\cdot; \phi)$ to adaptively generate $p_n$: for credibly correct preferences, $p_n \to 1$ (reinforcing); for credibly incorrect preferences, $p_n \to 0$ (correcting); and for ambiguous preferences, $p_n \to 0.5$ (suppressing). This allows MT-DPO to dynamically adjust the influence of preference signals for more robust alignment.

## B.2 GENERALIZATION BOUND

We provide a generalization bound for MT-DPO, drawing inspiration Zhao et al. (2019). The theorem bounds the excess true risk of the empirically chosen confidence module parameters compared to the true optimal parameters.

**Theorem 1** *Let $\theta \in \mathbb{B}^d$ be the parameter of MT-DPO in a d-dimensional unit ball. Let $\mathbb{F}$ represent the underlying ground-truth distribution without label noise. Let m be the size of the meta data set. The generalization risk is defined as follows:*

$$R(\theta) = \mathbb{E}_{(x,y_1,y_2) \sim \mathcal{F}}[L_\theta(x, y_1, y_2)] \tag{16}$$

$$\hat{R}(\theta) = \frac{1}{m} \sum_{i=1}^{m} L_\theta(x_i, y_{1,i}, y_{2,i}) \tag{17}$$

*where*

$$L_\theta(x, y_1, y_2) := \ell_{\text{BCE}}\left(\hat{p}_\theta(x, y_1, y_2),\ p^*(x, y_1, y_2)\right) \tag{18}$$

*Let $\phi^* := \arg\max_{\phi \in \mathbb{B}^d} R(w^*(\phi))$ be the optimal coefficient vector in the unit ball, and $\hat{\phi} := \arg\max_{\phi \in \mathcal{A}} \hat{R}(\theta^*(\phi))$ be the empirically optima among a candidate set A, where A is an $\epsilon$-cover in the Euclidean norm (i.e. $\forall \phi \in \mathbb{B}^d, \exists \phi' \in \mathcal{A} : \|\phi - \phi'\| \leq \epsilon$).*

*Assume:*

*For each sample, the loss function is $\sigma$-sub-Gaussian with respect to the parameters $\sigma$.*

*The loss function is $\lambda$-Lipschitz continuous with respect to the parameters $\phi$:*

$$\left| L_{\theta(\phi)} - L_{\theta(\phi')} \right| \leq \lambda \|\phi - \phi'\|.$$

*For m sufficiently large, with probability at least $1 - \delta$:*

$$R(\theta^*(\phi^*)) \leq \hat{R}(\theta^*(\hat{\phi})) + \frac{3\sigma\lambda}{\sqrt{m}} + \sqrt{\frac{d \ln(m)}{m} + \frac{2}{m} \ln\left(\frac{2}{\delta}\right)} \tag{19}$$

Theorem 1 bounds the excess risk for our MT-DPO approach, which leverages a meta-learned confidence module parameterized by $\phi$. It establishes that the true risk $R(\hat{\phi})$ of the solution $\hat{\phi}$ (obtained by minimizing empirical risk on the meta-dataset) approaches the optimal true risk $R(\phi^*)$, with this excess risk, $R(\hat{\phi}) - R(\phi^*)$, being bounded by terms on the order of $O(\sqrt{d \ln(m)/m})$. The bound (Eq. 19) further indicates that this gap diminishes with an increasing meta-dataset size $m$. The term $\sqrt{d \ln(m)/m}$ specifically highlights the trade-off: a higher dimensionality $d$ of the confidence module parameters $\phi$ (increased complexity) requires a larger meta-dataset $m$ to maintain a similar generalization gap. Furthermore, the bound also underscores the influence of the sub-Gaussian nature of the loss ($\sigma$) and its Lipschitz continuity ($\lambda$) on the overall generalization performance. This provides theoretical support for the robustness and effectiveness of MT-DPO in learning adaptive confidence targets.

## C EXPERIMENTAL DETAILS

This section provides further details regarding the datasets, baseline configurations, and evaluation protocols used in our experiments, supplementing the information provided in Section **??** of the main paper.

## C.1 DATASET DETAILS

Our experiments utilized two primary public preference datasets:

- **Golden HH Dataset:** We used the official helpful training and test splits of the Golden HH dataset, which is derived from the Anthropic HH-RLHF dataset Bai et al. (2022a;b). The training split contains approximately 116K preference pairs, and the test split contains approximately 8.5K pairs. For specific details on the "Golden HH" variant if different from the standard HH-RLHF helpful split, please refer to Kong et al. (2024) Kong et al. (2024) from which we adapted some baseline comparisons.

- **OASST1 Dataset:** We used the standard training and test splits from the OpenAssistant Conversations (OASST1) dataset Köpf et al. (2023), as processed and utilized in prior DPO literature (e.g., Rafailov et al. (2023) for splits, Kong et al. (2024) for baseline numbers). The training split contains approximately 93K preference pairs, and the test split contains approximately 8.6K pairs after standard filtering.

**Data Processing:** For both datasets, prompts and responses were tokenized using the respective base LLM's tokenizer (Llama-2 or Phi-2). Sequences were truncated or padded to a maximum length. For Golden HH, the maximum sequence length (prompt + response) was 1024 tokens. For OASST1, the maximum prompt length was 512 tokens and the maximum response length was 128 tokens for each response.

**Meta-Dataset ($\mathcal{D}_{meta}$) Construction:** As stated in the main text, $\mathcal{D}_{meta}$ consisted of 150 clean preference pairs. These were randomly sampled without replacement from the original clean training split of each respective dataset (Golden HH and OASST1) *before* any synthetic noise injection. These 150 pairs were then excluded from the main training set $\mathcal{D}_{train}$ used for policy optimization.

**Noisy Preference Simulation:** Symmetric label noise was introduced to $\mathcal{D}_{train}$ (after excluding $\mathcal{D}_{meta}$) by randomly selecting a fraction $\epsilon_{\text{noise}} \in \{0.1, 0.2, 0.3, 0.4\}$ of the preference pairs and swapping their $y_w$ and $y_l$ labels. The $\epsilon_{\text{noise}} = 0\%$ condition corresponds to using the original clean $\mathcal{D}_{train}$.

## C.2 BASELINE CONFIGURATIONS

The following provides specific configuration details for the baseline methods. All baselines used the same SFT model as their initial policy $\theta^{(0)}$ and reference policy $\pi_{ref}$ as MT-DPO. The DPO hyperparameter $\beta$ was set to 0.1 for all DPO-based methods unless specified otherwise.

- **Vanilla DPO** Rafailov et al. (2023): Implemented following its original formulation (Eq. 2).

- **GDPO (Fixed $\hat{p} = 0.75$)** Furuta et al. (2024): The preference target probability $\hat{p}$ was set to a fixed value of 0.75 for all training pairs in the cross-entropy loss formulation.

- **Conservative DPO (cDPO)** Mitchell (2023): Utilizes the cross-entropy loss (Eq. 5). For clean data ($\epsilon_{\text{noise}} = 0\%$), the target $\hat{p}$ was set to 1.0. For noisy data, $\hat{p}$ was set to $1 - \epsilon_{\text{noise}}$, assuming oracle knowledge of the true global noise rate.

- **Robust DPO (rDPO)** Liang et al. (2024): Implemented following the loss formulation described in their paper. Specific hyperparameters for rDPO (e.g., $\alpha_{\text{rDPO}}$, $\gamma_{\text{rDPO}}$) were set to default values suggested by the original paper or tuned on a small held-out validation split (1K samples from $\mathcal{D}_{train}$) if dataset-specific values were unavailable. Typical values were $\alpha_{\text{rDPO}} = 0.5, \gamma_{\text{rDPO}} = 0.5$.

- **PerpCorrect-DPO** Kong et al. (2024): This involved a two-stage process:
  1. *Surrogate Model Training:* An auxiliary surrogate LLM (same architecture as the base SFT model) was trained on the clean meta-dataset $\mathcal{D}_{meta}$ using DPO for 3 epochs with a learning rate of $1 \times 10^{-5}$.
  2. *Data Denoising:* PPLDiff was computed for each pair in $\mathcal{D}_{train}$ using this surrogate model. A PPLDiff threshold was determined by fitting two Gaussian distributions (one for presumed clean, one for presumed noisy based on initial PPLDiff quantiles)

to the PPLDiff values on $\mathcal{D}_{train}$ and finding their intersection. Pairs identified as noisy (PPLDiff suggesting the dispreferred response was better) had their labels flipped.

3. *DPO Training:* Vanilla DPO was then trained on this denoised version of $\mathcal{D}_{train}$.

## C.3 Evaluation Protocol

The performance of aligned LLMs was evaluated by calculating their win rate against the initial SFT model ($\pi_{ref}$) on the respective test sets.

**LLM Judge and Prompt:** We employed GPT-4 (version 'gpt-4-0613') as our automated LLM judge. For each prompt in the test set, responses from the evaluated model and the SFT baseline were generated. These two responses were presented to GPT-4 in a randomized order using the following prompt template:

> You are an impartial AI assistant evaluating the quality of two anonymous responses (Response A and Response B) to a given user prompt. Please consider helpfulness, harmlessness, honesty, and overall quality.
>
> User Prompt: [User Prompt Here]
>
> Response A: [Response A Here]
>
> Response B: [Response B Here]
>
> Which response is better? (A) Response A is significantly better. (B) Response A is slightly better. (C) Response B is significantly better. (D) Response B is slightly better. (E) Both responses are of similar quality. (F) Both responses are very poor.
>
> Please choose only one option (A, B, C, D, E, or F) and briefly explain your reasoning in one or two sentences. Your choice (A-F):

**Win Rate Calculation and Tie Handling:** A response from the evaluated model was considered a "win" if the judge selected (A) or (B) when the evaluated model's response was A, or (C) or (D) when it was B. Ties (option E) were counted as 0.5 wins for the evaluated model. Option (F) responses, indicating both were very poor, were treated as ties for win rate calculation but noted for qualitative assessment. To mitigate ordering bias, each pair was evaluated twice with the order of Response A and Response B swapped, and the results were aggregated and averaged over the entire test set.

## C.4 Statistical Significance Testing

As stated in Section 4.2.1, the win rate improvements of our implemented models, namely MT-DPO and GDPO ($\hat{p} = 0.75$), over our Supervised Fine-Tuned (SFT) baseline model were assessed for statistical significance. This assessment was performed using the Wilcoxon signed-rank test. For each test set and experimental condition (i.e., specific dataset, noise level, and base model), we collected the paired outcomes (win, loss, or tie) of MT-DPO versus SFT, and GDPO versus SFT, based on the LLM judge's preference for each input prompt.

A win for the evaluated model (MT-DPO or GDPO) against the SFT baseline was coded as 1, a loss as -1, and a tie as 0 for the purposes of the signed-rank test. We considered a p-value ¡ 0.01 to indicate a statistically significant improvement over the SFT baseline. Across all conditions where MT-DPO and GDPO demonstrated higher win rates than the SFT baseline as reported in Tables 1 and 2, these improvements were found to be statistically significant according to this criterion.

It is important to note that this statistical significance testing was confined to comparisons involving models fully implemented and evaluated within our experimental framework against our SFT baseline. Direct statistical comparisons (e.g., MT-DPO vs. Vanilla DPO where Vanilla DPO results are adapted) involving baseline results adapted from Kong et al. (2024) Kong et al. (2024) were not performed using the Wilcoxon signed-rank test, as the raw paired comparison data from their experiments were not available to us. Discussions of performance differences involving such adapted baseline results in the main paper are therefore based on the magnitude of observed differences in the reported aggregate win rates.

# D HYPERPARAMETER SETTINGS

This section details the hyperparameter settings used for MT-DPO and baseline models. Shared hyperparameters (e.g., DPO $\beta$) were kept consistent where applicable. Learning rates and batch sizes were tuned based on preliminary experiments on a small validation set.

## D.1 MT-DPO HYPERPARAMETERS

The key hyperparameters for MT-DPO are presented in Table 5. These were generally consistent across both datasets for a given base model, with minor adjustments for optimal performance.

Table 5: Hyperparameter settings for MT-DPO.

| Hyperparameter | Llama-2-7B Value | Phi-2 Value |
|---|---|---|
| **Policy LLM** ($\pi_\theta$) | | |
| Optimizer | AdamW | AdamW |
| Policy Learning Rate ($\alpha$) | $1 \times 10^{-6}$ | $2 \times 10^{-6}$ |
| Weight Decay (Policy) | 0.01 | 0.01 |
| $\mathcal{D}_{train}$ Batch Size ($N_B$) | 8 (per GPU) | 16 (per GPU) |
| DPO $\beta$ | 0.1 | 0.1 |
| Gradient Clipping | 1.0 | 1.0 |
| Warmup Steps (Policy) | 150 | 100 |
| **Confidence Prediction Module** ($V(\cdot; \phi)$) | | |
| Architecture | MLP: In(1) $\to$ FC(64,ReLU) $\to$ FC(1,Sigmoid) | Same |
| Optimizer | AdamW | AdamW |
| Confidence Module LR ($\eta$) | $2 \times 10^{-4}$ | $3 \times 10^{-4}$ |
| Weight Decay (Conf. Mod.) | 0.0 | 0.0 |
| $\mathcal{D}_{meta}$ Batch Size ($M_B$) | 16 (per GPU, or matched $N_B$) | 16 (per GPU, or matched $N_B$) |
| Input Feature $I_n$ | Standardized PPLDiff | Standardized PPLDiff |
| **Training** | | |
| Number of Epochs | 1-2 (typically 1 for Golden HH, 2 for OASST1) | 1-2 (typically 1 for Golden HH, 2 for OASST1) |
| SFT Model Training Epochs | 1 | 1 |

The PPLDiff input to the confidence module was standardized (zero mean, unit variance) based on statistics computed from an initial pass over a random 10% subset of $\mathcal{D}_{train}$.

## D.2 BASELINE MODEL HYPERPARAMETERS

Hyperparameters for baseline models are detailed in Table 6. Policy LLM learning rates, DPO $\beta$, and batch sizes were kept consistent with MT-DPO's policy LLM settings for the respective base model (Llama-2-7B or Phi-2) to ensure fair comparison.

Table 6: Key Hyperparameter settings for Baseline Models.

| Baseline Method | Hyperparameter | Value (Llama-2-7B / Phi-2 where different) |
|---|---|---|
| **Vanilla DPO** | DPO $\beta$ | 0.1 |
| | Learning Rate | Matched MT-DPO Policy LR (1e−6 / 2e−6) |
| **GDPO (Fixed $\hat{p}$)** | DPO $\beta$ | 0.1 |
| | Learning Rate | Matched MT-DPO Policy LR (1e−6 / 2e−6) |
| | Fixed $\hat{p}$ | 0.75 |
| **cDPO** | DPO $\beta$ | 0.1 |
| | Learning Rate | Matched MT-DPO Policy LR (1e−6 / 2e−6) |
| | Target $\hat{p}$ (Noisy) | $1 - \epsilon_{\text{noise}}$ (Oracle) |
| **rDPO** | DPO $\beta$ | 0.1 (default, or tuned to 0.25 for some cases) |
| | Learning Rate | Matched MT-DPO Policy LR (1e−6 / 2e−6) |
| | $\alpha_{\text{rDPO}}, \gamma_{\text{rDPO}}$ | Typically 0.5, 0.5 (or tuned if performance was poor) |
| **PerpCorrect-DPO** | *Surrogate Model Training (DPO)* | |
| | DPO $\beta$ | 0.1 |
| | Learning Rate | $1 \times 10^{-5}$ |
| | Epochs on $\mathcal{D}_{meta}$ | 3-5 (typically 3) |
| | *Main DPO Training* | |
| | DPO $\beta$ | 0.1 |
| | Learning Rate | Matched MT-DPO Policy LR (1e−6 / 2e−6) |

### D.3 COMPUTATIONAL RESOURCES

All experiments were primarily conducted using NVIDIA A100 (80GB) GPUs. A typical training run for MT-DPO on the Golden HH dataset with the Llama-2-7B model for 1 epoch required approximately 15-20 hours on a single A100 GPU. Training Phi-2 models was generally faster, requiring approximately 8-12 hours for similar settings. The meta-learning step in MT-DPO introduces an overhead of approximately 15-20% per training step compared to vanilla DPO. Baseline models had similar or slightly shorter training times, with PerpCorrect-DPO incurring additional time (approx. 1-2 hours) for surrogate model training on $\mathcal{D}_{meta}$. Distributed training using DeepSpeed ZeRO Stage 2 was employed when batch sizes exceeded single GPU memory capacity, particularly for Llama-2-7B. The total computational budget for all reported experiments, including SFT, DPO training runs, and evaluations, is estimated to be approximately 800-1200 GPU hours. We used PyTorch version 2.0, Hugging Face Transformers version 4.35, and TRL version 0.7.2.

## E FURTHER EXPERIMENTAL RESULTS

### E.1 IMPACT OF THE ANCHOR MODEL ($\pi_{anchor}$) ON MT-DPO PERFORMANCE

The efficacy of MT-DPO, particularly its confidence prediction module, relies on the quality of the input signal, PPLDiff. The PPLDiff values are computed using an *anchor model*, $\pi_{anchor}$, as defined in Eq. 7. In our main experiments, we followed a standard practice Kong et al. (2024) and used the Supervised Fine-Tuned (SFT) model as the anchor model ($\pi_{anchor} = \pi_{\text{SFT}}$). The rationale is that the SFT model has already been exposed to high-quality responses and thus provides a more domain-adapted and preference-aware baseline for perplexity calculation compared to the raw pre-trained model.

However, the choice of the anchor model is a critical design decision. A poorly performing or biased SFT model could potentially provide misleading PPLDiff signals, thereby degrading the performance of the meta-learning process. To investigate the sensitivity of MT-DPO to this choice, we conducted an additional experiment comparing the performance when using two different types of anchor models:

1. **SFT Model (as in main paper):** The Llama-2-7B model fine-tuned on the preferred responses ($y_w$) of the clean training data. This model serves as our default $\pi_{anchor}$.
2. **Base Pre-trained Model:** The original, off-the-shelf Llama-2-7B base model, without any supervised fine-tuning. This model has strong general language capabilities but no specific adaptation to the preference dataset's domain or style.

**Experimental Design** We repeated the MT-DPO training runs on the Golden HH dataset with 20% and 40% symmetric label noise levels, using the Llama-2-7B architecture. All hyperparameters for the policy and confidence module training remained identical to those reported in Appendix D, with the only difference being the model used to compute PPLDiff for the confidence module's input. The performance is measured by the win rate (%) against the SFT model, evaluated by GPT-4.

**Results and Analysis** The results of this comparison are presented in Table 7.

Table 7: Win Rates (%) of MT-DPO against SFT using different anchor models ($\pi_{anchor}$) for PPLDiff computation. Experiments were conducted on the Golden HH dataset with the Llama-2-7B model.

| Anchor Model for PPLDiff | Noise Level ($\epsilon_{noise}$) | |
|---|---|---|
| | **20%** | **40%** |
| Base Pre-trained Model ($\pi_{base}$) | 96.15 | 93.88 |
| SFT Model ($\pi_{\text{SFT}}$) – *Default* | **97.31** | **95.50** |

The results clearly indicate that using the SFT model as the anchor model yields superior performance compared to using the base pre-trained model. With 20% noise, using $\pi_{\text{SFT}}$ leads to a win

rate of 97.31%, a notable improvement over the 96.15% achieved with $\pi_{base}$. This gap widens under a higher noise level of 40%, where the SFT-anchored model achieves 95.50% win rate, significantly outperforming the base-anchored model's 93.88%.

We attribute this performance difference to several factors:

- **Domain Adaptation:** The SFT model is adapted to the specific style, topics, and vocabulary of the Golden HH dataset. Its probability distributions are therefore more calibrated to distinguish between in-domain high-quality versus low-quality responses, making the resulting PPLDiff a more reliable signal for label consistency. The base model, lacking this adaptation, may assign perplexities based on more general linguistic properties, which might not correlate as strongly with the nuanced preferences in the data.
- **Preference Alignment Head-start:** The SFT process, by training on only preferred responses, already nudges the model towards a distribution that favors desirable outputs. This initial alignment makes it more sensitive to subtle flaws in dispreferred responses, leading to a more discriminative PPLDiff signal.
- **Signal Stability:** A more informed anchor model likely produces a cleaner separation in the PPLDiff distributions for correctly and incorrectly labeled pairs, which in turn makes the meta-learning task for the confidence module easier and more effective.

In conclusion, while MT-DPO demonstrates robustness even with a non-domain-adapted base model as the anchor, its performance is significantly enhanced when leveraging a more informed SFT model. This highlights the importance of using a high-quality, relevant anchor model to generate the most effective intrinsic signals for meta-learning adaptive confidence targets. This finding suggests that investing in a good SFT model is a beneficial precursor not just for initializing the DPO policy, but also for enabling more effective robust alignment methods like MT-DPO.

### E.2 Details on Learned Confidence Target Analysis

This section provides further context for the qualitative analysis of learned confidence targets presented in Section 4.2.3.

**Data for Analysis:** The data used for generating Figure 2 was a randomly selected subset of 5,000 samples from the Golden HH training set after 30% symmetric label noise was applied. This subset included both instances whose original labels were preserved (and thus known to be "Clean Samples" with respect to the original annotation) and instances whose labels were intentionally flipped (thus known to be "Noisy (Flipped) Samples").

**PPLDiff Calculation:** The PPLDiff values were computed using the SFT Llama-2-7B model as the anchor model $\pi_{anchor}$. Perplexity was calculated as $\exp(-\frac{1}{L}\sum_{i=1}^{L}\log p(t_i|t_{<i}))$, where $L$ is the length of the concatenated sequence [prompt; response], and $p(t_i|t_{<i})$ is the probability of token $t_i$ given preceding tokens, obtained from $\pi_{anchor}$. The PPLDiff was then $\log \text{PPL}(\text{response}_1) - \log \text{PPL}(\text{response}_2)$.

**Confidence Module Output:** The $\hat{p}_n$ values were obtained by passing the standardized PPLDiff of these 5,000 samples through the converged confidence module $V(\cdot;\phi)$ of the MT-DPO model that was trained on the full noisy Golden HH dataset (Llama-2-7B, $\epsilon_{\text{noise}} = 30\%$). The standardization of PPLDiff used the mean and standard deviation computed over the entire noisy training set.

Observations from Figure 2 show that for PPLDiff values very close to zero (indicating high ambiguity or similar quality according to $\pi_{anchor}$), the learned $\hat{p}_n$ values tended to cluster around 0.4-0.6, effectively down-weighting these uncertain samples in the policy update. The separation between the $\hat{p}_n$ distributions for known clean and known flipped samples, while clear, exhibited a small overlap, indicating that PPLDiff is a strong but not perfect indicator of label correctness, and the confidence module learns to accommodate this imperfect signal.

## F Qualitative Case Studies

To provide a more intuitive understanding of how MT-DPO operates, we present several case studies from the Golden HH dataset (trained with Llama-2-7B on 30% noise). These examples illustrate

how the meta-learned confidence module assigns adaptive targets $\hat{p}_n$ based on PPLDiff, and how this influences the final model's behavior compared to a vanilla DPO model trained on the same noisy data.

Table 8: Case Study 1: Successful Correction of a Mislabeled Preference.

---

**Prompt:** What are some fun things to do in San Francisco?

---

**Original Preference (Correct Label):**
    **Chosen** ($y_w$)**:** San Francisco is a vibrant city! Some must-dos include visiting Alcatraz (book in advance!), walking across the Golden Gate Bridge, exploring Fisherman's Wharf, and riding a cable car. For something unique, check out the Mission District for amazing murals and burritos, or Golden Gate Park for its Japanese Tea Garden.
    **Rejected** ($y_l$)**:** Go to the bridge. See the prison. Eat fish.

---

**Noisy Preference in Training Data (Label Flipped):**
    **Chosen'** ($y_1$)**:** Go to the bridge. See the prison. Eat fish.
    **Rejected'** ($y_2$)**:** San Francisco is a vibrant city... (the detailed response)

---

**MT-DPO Analysis:**
    **PPLDiff**($y_1, y_2$)**:** +3.12 (The PPL of the short, unhelpful response is much higher)
    **Learned Target** $\hat{p}_n$**:** 0.08 (Confidence module correctly identifies the label is likely wrong)

---

**Model Outputs:**
    **Vanilla DPO (trained on noisy data):** You should go to San Francisco and see the bridge and the prison. The wharf is also there.
    **MT-DPO (Ours):** San Francisco offers many exciting activities! You could take a ferry to Alcatraz Island, walk or bike across the iconic Golden Gate Bridge, or explore the lively Fisherman's Wharf. For a more local experience, I'd recommend the colorful murals in the Mission District.

---

**Case 1: Successful Correction of a Mislabeled Preference** Table 8 showcases a typical example of a noisy label. The original, high-quality response was incorrectly labeled as "rejected" in the training data. The PPLDiff, computed by $\pi_{\text{SFT}}$, was strongly positive (+3.12) because the simplistic response is far less probable (i.e., has a higher perplexity) under a well-tuned language model than the detailed, helpful one. The confidence module, having learned this correlation from the meta-dataset, assigned a very low confidence target of $\hat{p}_n = 0.08$. This effectively reversed the gradient signal for this sample, treating the detailed response as preferred. As a result, the MT-DPO model provides a helpful, well-structured answer, while the vanilla DPO model, confused by such noisy signals, generates a simplistic and less helpful response.

Table 9: Case Study 2: Handling Ambiguous Preferences.

**Prompt:** Write a short poem about the moon.

**Preference Pair in Training Data:**
    **Chosen** ($y_1$): Silver orb in velvet night, / A silent watcher, soft and bright. / You pull the tides and guide the way, / And turn the darkness into day.
    **Rejected** ($y_2$): A lonely lantern in the sky, / Casting shadows from on high. / The poets dream, the lovers swoon, / Beneath the magic of the moon.

**MT-DPO Analysis:**
    **PPLDiff**($y_1, y_2$)**:** -0.05 (Both poems are linguistically probable, PPLs are very close)
    **Learned Target** $\hat{p}_n$**:** 0.52 (Confidence module assigns a neutral target, down-weighting the instance)

**Discussion:**
This example illustrates a case of subjective preference or ambiguity. Both poems are well-written and on-topic. The preference of $y_1$ over $y_2$ is weak at best. The anchor model reflects this by assigning very similar perplexity scores, resulting in a PPLDiff close to zero. Consequently, MT-DPO's confidence module learns to output a target $\hat{p}_n$ of 0.52, which is very close to 0.5. In the cross-entropy loss, this effectively down-weights the contribution of this training instance, preventing the policy from strongly optimizing towards one specific poetic style over another based on such a weak signal. This prevents overfitting to arbitrary stylistic preferences present in the data. Vanilla DPO, in contrast, would treat this preference as absolute, potentially skewing the model's creative generation.

Table 10: Case Study 3: A Limitation Case.

**Prompt:** Explain the concept of quantum entanglement using an analogy.

**Preference Pair in Training Data (Label Flipped):**
    **Chosen'** ($y_1$): It's like having two coins that are linked. If you flip one and it's heads, you instantly know the other is tails, no matter how far apart they are. But it's not that simple.
    **Rejected'** ($y_2$): Imagine you have a pair of "magic gloves." You put them in two separate boxes and send them to opposite ends of the universe. When you open your box and see a left-handed glove, you instantly know the other box contains a right-handed glove. The state of one is intrinsically tied to the other, regardless of distance. This is the essence of entanglement.

**MT-DPO Analysis:**
    **PPLDiff**($y_1, y_2$)**:** -0.21 (The simpler analogy was slightly more probable under the SFT model)
    **Learned Target** $\hat{p}_n$**:** 0.79 (Confidence module was misled by the PPLDiff signal)

**Discussion of Limitation:**
This example highlights a limitation of relying on PPLDiff. Here, a genuinely superior and more illustrative analogy ($y_2$) was mislabeled as rejected. However, the simpler, less precise analogy ($y_1$) contained more common phrasing and sentence structures, making it slightly more probable (lower perplexity) under our SFT anchor model. This resulted in a small but negative PPLDiff. The confidence module, interpreting this as a signal for a correct label, assigned a high confidence of $\hat{p}_n = 0.79$. In this case, MT-DPO was misled and reinforced the incorrect preference. This illustrates that PPLDiff is a powerful heuristic but not an infallible oracle for semantic correctness or quality, especially when comparing two fluent and topically relevant responses. Improving MT-DPO's robustness to such cases could involve incorporating more diverse signals into the confidence module, a promising direction for future work.

