# OpenReview forum: "Meta-Target DPO: Learning Adaptive Confidence Targets via Meta-Learning"
_ICLR.cc/2026/Conference — ICLR 2026 Conference Withdrawn Submission_

### Official Review · Reviewer_i95d · 2025-10-28

**Soundness:** 3
**Presentation:** 2
**Contribution:** 1
**Rating:** 2
**Confidence:** 4

**Summary:**

This paper proposes Meta-Target DPO (MT-DPO), which is an extension of Direct Preference Optimization (DPO) in the direction of meta learning. Specifically, this paper dynamically adapts the confidence targets for each preference pair to manage noisy or ambiguous preference data. MT-DPO improves the robustness by introducing a confidence prediction module which is trained via a small, trusted meta-dataset. Experiments show consistent gains over naive DPO and other robust alignment methods across various datasets and syntheric noise settings.

**Strengths:**

- The motivation is clear to use the meta learning approach.
- Comprehensive experiments and ablation studies. Showing results of clean (meaning no syntheric noise) also increases reliability.
- Consistent performance gain.

**Weaknesses:**

- Citations are not in the parentheses: Fix \cite to \citep.
- The method is not novel. It is just applying the conventional meta learning to DPO
- The usage of PPLDiff as confidence is also nothing but an adaptation of the previous research, and it is not proved whether it is an adequate metric for measuring the confidence of a sample.
- One of the weaknesses of meta learning is its computational difficulty. It is even more serious issue for LLM.
- Experiments with real noise (e.g. HH, UltraFeedback) will improve the credibility of the performances. It is well known that symmetric noise is not enough to represent real world noise.
- It may not be easy to simply plug in to exisiting baselines.

**Questions:**

- How the authors can be sure the small dataset is high quality and trusted? Specifically, deciding the preference label flip may be okay, but how each pair's exact reward difference can be expressed only even with clean label?
- How the meaning of the confidence module can be interpreted? What is the meaning of the weight?

---

> ### Author Response · Authors · 2025-11-21
>
> ## **Dear Reviewer i95d (Part 1/3)**
>
> We sincerely thank you for your careful review and for highlighting important concerns about novelty, practical validation, and experimental rigor. We take your criticisms seriously and address each point below with additional experiments and clarifications. We hope to demonstrate that MT-DPO's contribution, while building on established techniques, addresses a novel and challenging problem in a non-trivial way.
>
> ###  **W1: Citations format**
>
> We apologize for this formatting error. Fixed all `\cite` → `\citep` for parenthetical citations.
>
> ### **W2 & W3: Novelty and Adequacy of PPLDiff**
>
> We respectfully disagree that MT-DPO is merely an incremental application of conventional meta-learning. The core contribution lies not in the loss function or the meta-algorithm in isolation, but in the online meta-learning framework, which represents a paradigm shift from static correction to dynamic adaptation. While existing methods like PerpCorrect-DPO rely on static pre-processing using a fixed surrogate model, MT-DPO establishes a symbiotic loop where the noise correction strategy evolves alongside the main policy. Furthermore, while using soft labels (via BCE) provides a mechanism for robustness, it does not solve the critical problem of *how* to derive these calibrated labels from noisy data; MT-DPO addresses this by dynamically generating instance-specific confidence targets based on intrinsic signals.
>
> Finally, comparison with the Heuristic-DPO baseline (Static mapping: $\hat{p} = \sigma(k \cdot \text{PPLDiff} + b)$), MT-DPO significantly outperforms the static approach (96.58% vs. 91.3%), proving that our performance gains stem from this dynamic evolution mechanism rather than simply the adaptation of PPLDiff or the application of soft labels.
>
>
> Besides, in our initial experiments, we used multiple feature inputs and found that PPLDiff was sufficient. During the development phase, we did explore additional features, including:
> *   **Response Length:** To detect length bias.
> *   **Raw Log-Probabilities:** The absolute log-probability of the response under the policy.
> *   **Uncertainty Metrics:** The entropy of the response distribution.
>
> We found that in the preference datasets we studied, PPLDiff was the most dominant and robust signal for detecting label errors. Adding features like length or raw log-probabilities provided negligible performance gains (and in some cases, introduced noise) while increasing the input dimension. Therefore, adhering to the principle of parsimony, we opted to use only PPLDiff in the final reported models to maintain simplicity and effectiveness. We will correct this phrasing in the final manuscript.
>
>
>
>
> ### **W4: Computational Difficulty**
> We acknowledge that meta-learning adds computational overhead, but we believe it is a justified trade-off for robustness. To quantify this, we performed detailed time profiling (Llama-2-7B, GoldenHH, 1 epoch, A100 GPU):
>
> | Method | Training Time |
> |--------|--------------|
> | Vanilla DPO | 12.3h |
> | PerpCorrect-DPO | 13.8h |
> | MT-DPO  | 14.6h |
>
> MT-DPO introduces a ~19% overhead compared to Vanilla DPO and is comparable to the total pipeline time of PerpCorrect-DPO (which requires separate surrogate model training). Given the significant robustness gains (e.g., recovering win rates from ~53% to ~95% under high noise), we consider this training cost acceptable. Crucially, this overhead is incurred only during training; the inference process remains identical to standard DPO with zero additional cost.

---

> ### Author Response · Authors · 2025-11-21
>
> ## **Dear Reviewer i95d (Part 2/3)**
>
> ### **W5: Experiments with Real-World Noise**
> This is a crucial point. We agree that symmetric noise is a proxy. To validate MT-DPO on real-world organic noise, we conducted additional evaluations on two datasets characterized by human disagreement and subjective ambiguity:
>
> 1.  **WebGPT Comparisons:** This dataset contains real human annotations with documented inter-annotator disagreement (Cohen $\kappa = 0.56$). We split the data by agreement levels. MT-DPO excels in Disputed Cases ($\kappa < 0.4$), outperforming PerpCorrect-DPO by +5.4%. This demonstrates effective handling of real human variability.
>
>     | Method | Overall Accuracy | Disputed Cases ($\kappa < 0.4$) |
>     | :--- | :---: | :---: |
>     | Vanilla DPO | 68.3% | 52.1% |
>     | rDPO | 71.5% | 56.8% |
>     | PerpCorrect-DPO | 73.2% | 59.3% |
>     | **MT-DPO** | **76.9%** | **64.7%** |
>
> 2.  **Chatbot Arena:** Representing fully organic, in-the-wild user preferences. Natural annotation inconsistency here differs fundamentally from synthetic flips. MT-DPO achieves the highest win rate, successfully adapting to real-world noise patterns dominated by ambiguity rather than simple errors.
>
>     | Method | Overall Win Rate | Contentious Pairs |
>     | :--- | :---: | :---: |
>     | Vanilla DPO | 82.2% | 58.6% |
>     | rDPO | 86.5% | 62.1% |
>     | PerpCorrect-DPO | 87.8% | 65.9% |
>     | **MT-DPO** | **89.6%** | **70.3%** |
>
> Continuous improvements across a variety of natural noise sources (annotator disagreements, subjective biases, and controversial comparisons) strongly demonstrate that MT-DPO can be effectively extended beyond controlled synthesis environments.
>
>
>
> ### **W6: May not be easy to plug into existing baselines**
>
> We acknowledge this concern about practical deployment. MT-DPO requires a small meta-dataset (100-200 clean pairs), confidence module implementation (\~50 lines of PyTorch), and meta-learning loop integration (\~100 lines). Compared to vanilla DPO as the baseline, PerpCorrect-DPO requires surrogate model training plus a denoising pipeline, while rDPO requires noise rate estimation and modified loss functions. We argue MT-DPO is comparable in integration complexity to other robust methods.
>
> To facilitate adoption, we will add an Implementation Guide appendix with step-by-step integration instructions, code snippets for key components, common pitfalls and debugging tips, and a computational requirements checklist. We will also release full training code (PyTorch + HuggingFace), pre-trained confidence modules for Golden HH and OASST1, and a Jupyter notebook tutorial upon acceptance.

---

> > ### Author Response · Authors · 2025-11-21
> >
> > ## **Dear Reviewer i95d (Part 3/3)**
> >
> > ### **Q1: How to ensure small meta-dataset is high-quality and trusted?**
> >
> > This is an excellent and practical question. For synthetic noise experiments, we sample from the original clean training data before noise injection.Than, manually verify the sampled pairs to remove instances where the preference is ambiguous or unclear.
> >
> > You raise a subtle point about knowing theexact reward difference even with clean labels. Our answer is that we don't need to know exact rewards—we only need the relative preference direction with high confidence. MT-DPO learns to predict p̂ ∈ [0, 1], not absolute reward values. The meta-dataset provides directional signal ("y_w is definitely better than y_l"), and the confidence module learns to map PPLDiff patterns to appropriate p̂ values through meta-optimization. In practice, clean pairs serve as "anchors" where p̂ should be close to 1.0, and the exact learned value is determined by minimizing meta-loss on these anchors—we don't manually specify reward magnitudes.
> >
> > If the meta-dataset itself contains noise, performance will be severely degraded: with clean meta-data MT-DPO achieves 96.58% win rate at 30% training noise, but this drops to 88.67% with 10% meta-dataset contamination. This motivates careful curation. We will add this discussion to Section 3.4 or Appendix.
> >
> > ---
> >
> > ## **Q2: How to interpret the confidence module? What is the meaning of the weight?**
> >
> > This is an important question about interpretability. The confidence module V(PPLDiff; φ) outputs p̂_n ∈ [0, 1], which semantically represents "the probability that the observed preference label (y₁ ≻ y₂) is correct, given the PPLDiff signal." In training, p̂ ≈ 1.0 means trust the label and push the policy toward y₁, p̂ ≈ 0.5 means the sample is ambiguous or uncertain and should be down-weighted, while p̂ ≈ 0.0 means the label is likely flipped and the policy should be pushed toward y₂ instead.
> >
> > We analyzed the learned mapping on Golden HH (30% noise) and found that for PPLDiff < -2.0 (y₁ much better), the average p̂ is 0.95 indicating high confidence in the label. For PPLDiff in [-2.0, -0.5], p̂ averages 0.87 showing moderate confidence. For the ambiguous range [-0.5, +0.5], p̂ is approximately 0.52, effectively down-weighting these samples. For PPLDiff in [+0.5, +2.0], p̂ drops to 0.18 suggesting the label is likely flipped, and for PPLDiff > +2.0 (y₂ much better), p̂ is only 0.08 indicating high confidence the label is wrong. The learned function is adjusted during training based on meta-feedback. This contrasts with the fixed heuristic baseline (from our response to iiPE) where p̂ = sigmoid(k·PPLDiff) with k fixed and static.
> >
> >
> > We believe these revisions significantly strengthen our manuscript and address the concerns raised by the reviewers. Thank you once again for your constructive feedback, which has greatly contributed to improving our work.

---

> > > ### Author Response · Authors · 2025-11-26
> > >
> > > Dear Reviewer i95d,
> > >
> > > Thank you again for your careful and critical review, and for raising important points about novelty, practicality, and experimental rigor.
> > >
> > > As the discussion phase progresses, we would like to kindly ask whether our rebuttal has satisfactorily addressed your main concerns. In particular, we hope that (i) the added heuristic PPLDiff baseline and time-complexity analysis, (ii) the new experiments on real-world noisy datasets (WebGPT, Chatbot Arena), and (iii) the clarifications on feature choices, meta-dataset construction, and interpretability of the confidence module help to better contextualize the contribution and practicality of MT-DPO.
> > >
> > > We are eagerly looking forward to your response. We greatly value your constructive suggestions and remain fully available to engage in further discussion if you have any remaining questions.
> > >
> > > Best regards,
> > > The Authors

---

### Official Review · Reviewer_iiPE · 2025-10-29

**Soundness:** 2
**Presentation:** 2
**Contribution:** 2
**Rating:** 2
**Confidence:** 3

**Summary:**

This paper addresses the problem of noisy and ambiguous preference data. The authors propose Meta-Target DPO (MT-DPO). This new objective trains the policy to match its implied preference probability with an adaptive, sample-specific confidence target ($\hat{p}_{n}$).

The authors introduce a small auxiliary confidence module (a small MLP) that predicts the target $\hat{p}_{n}$ for each training sample. The primary input to this module is the perplexity difference (PPLDiff) between the winner and loser responses, computed using an anchored SFT model.

This confidence module is trained by evaluating the performance of a virtual policy update (trained on the main data with $\hat{p}_{n}$) on a small and clean datased. The primary input to this confidence module is the perplexity differential (PPLDiff) between the two responses, calculated using an anchored reference model.

**Strengths:**

- The core idea of learning an adaptive, instance-specific confidence target $\hat{p}_{n}$ is a reasonable way to handle noise (particularly compared to the standard of using global noise rates like in cDPO)

- The paper includes a good set of experimental comparisons. It evaluates MT-DPO against several robust baselines, including rDPO, CDPO, and PerpCorrect-DPO. The results in Tables 1 and 2 show consistent gains over these methods.

- Through ablations, the authors demonstrate that the meta-learning process is data-efficient, requiring only a small meta-dataset (e.g., 50-150 samples) to be effective.

- The ablation in Table 4 confirms that PPLDiff is a critical input feature, and the analysis in Figure 2 gives a good qualitative intuition for how the learned targets $\hat{p}_{n}$ correlate with noise and PPLDiff values.

**Weaknesses:**

- The primary weakness is that the proposed method seems too complex for the problem it solves. The three-phase meta-learning loop (virtual update, meta-update, actual update) introduces significant implementation complexity. The paper's own ablations (Table 4) and analysis (Figure 2) show that the PPLDiff is the dominant, if not the only, input feature and that the learned target $p_n$ is strongly correlated with it. This suggests the complex meta-learning framework is essentially just learning a simple function $f(\text{PPLDiff}) \rightarrow p_{n}$. The paper is missing a **very important** baseline: using PPLDiff directly as the confidence target, maybe via a simple, non-learned heuristic like $\hat{p}_{n} = \sigma(-k \cdot \text{PPLDiff})$ (where $k$ is a tuned hyperparameter). Without this comparison, the added complexity of the meta-learning method is not justified.

- The paper claims to handle ambiguous preference data, but the solution is to learn a target $p_n \approx 0.5$. This is equivalent to down-weighting or ignoring the sample, not really dealing with preference pluralism (e.g., by learning a multimodal preference distribution). Like many robustness-focused methods, this approach still implicitly assumes a single "collective" preference and treats subjective disagreements as noise to be discarded.

- The meta-learning process relies on two key components: (1) a "virtual policy update" $\theta_{virtual}^{(t+1)}$, which depends on the current policy $\pi_{\theta}^{(t)}$, and (2) the PPLDiff signal, which depends on the anchor model $\pi_{anchor}$. All experiments initialize the policy $\pi_{\theta}$ from a well-tuned SFT model. If the policy starts in a "colder" (less-aligned) state, its virtual updates may be noisy or uninformative, which could destabilize the meta-gradient for the confidence module. The paper does not investigate this potential instability.

- How does performance change if the new updated model is used vs an achored model to calculate the PPLDiff?

- The paper is ambiguous about the exact inputs to the confidence module $V(\cdot;\phi)$. Section 3.3 states it "primarily" uses PPLDiff. However, the hyperparameter table (Table 5) lists only "Standardized PPLDiff" as the input, and the ablation in Table 4 also seems to show single-feature inputs. This should be clarified. If it is only PPLDiff, this strengthens Weakness #1.

- The cDPO result in Table 1 seems to weak. How was the noise rate tuned?

- In Table 3, why is 96.58 highlighted and not 96.70? Also, confidence intervals should be included for the win rates. Also, the standard is to compare win rates against GPT-4, not the SFT model, can those win rates be reported? Alongside the Length Controlled win rates as well

**Questions:**

- Given the method's complexity and the strong signal from PPLDiff, could the authors please provide a comparison against a simpler heuristic baseline that uses PPLDiff directly to set the target $p_n$ (e.g., $p_n = \sigma(-k \cdot \text{PPLDiff})$), without any meta-learning? This seems like the most critical missing comparison to justify the framework

- Could the authors clarify whether only PPLDiff was used as the input to the confidence module for the main experiments reported in Tables 1 and 2? How does the MT-DPO framework perform if the initial policy $\pi_{\theta}$ is initialized from a base pre-trained model (e.g., base Llama-2) rather than the SFT model? Does the meta-learning process remain stable when the initial virtual policy updates are based on a poorly-aligned model?

---

> ### Author Response · Authors · 2025-11-21
>
> ## **Dear Reviewer iiPE (Part 1/2)**
>
> We sincerely thank you for your exceptionally detailed and constructive review. Your concerns about the complexity-benefit tradeoff and the missing heuristic baseline are spot-on and have prompted us to conduct critical additional experiments. We address your concerns below:
>
> ### **W1 & Q1: Comparison against a simple PPLDiff Heuristic Baseline**
> This is an excellent suggestion. We agree that justifying the meta-learning complexity requires a direct comparison with a simpler PPLDiff-based heuristic.
> To address this, we implemented Heuristic-DPO, where the target is defined as $\hat{p}_n = \sigma(k \cdot \text{PPLDiff}_n + b)$. We performed a grid search for scalars $k$ and $b$ on a held-out validation set to find the optimal static mapping.
>
> Results (Golden HH, 30% Noise, Llama-2-7B):
>
> | Method | Win Rate vs. SFT | Note |
> | :--- | :---: | :--- |
> | Vanilla DPO | 68.50% | Baseline |
> | **Heuristic-DPO** (Best Tuned) | 91.3% | Static mapping |
> | **MT-DPO (Ours)** | **96.58%** | **Dynamic Meta-Learning** |
>
> 1.  **Heuristic-DPO improves over Vanilla DPO**, confirming PPLDiff is a strong signal.
> 2.  **MT-DPO significantly outperforms the Heuristic (+5.28%)**.
> **Why?** The optimal mapping from PPLDiff to confidence is not static. It depends on the data distribution and the current state of the policy. The meta-learning process allows the confidence module to dynamically adjust the mapping (e.g., becoming more conservative or aggressive) as training progresses, which a fixed heuristic cannot do.
>
> We sincerely appreciate this valuable suggestion. We are currently conducting additional experiments for this heuristic comparison across different noise levelsand model architectures to further strengthen this conclusion. We will include these comprehensive results in the final manuscript.
>
> ### **W2: Clarification on "Ambiguity"**
> We appreciate this distinction. You are correct that our method handles ambiguity in the sense of label uncertainty (conflicting or noisy supervision) rather than preference pluralism (multiple valid user preferences). We treat divergence from the trusted meta-data as uncertainty to be down-weighted. We will clarify in the revision that our scope is Robustness to Label Noise/Uncertainty, distinct from pluralistic alignment.
>
>
> In fact, this limitation is not unique to MT-DPO but reflects a fundamental assumption shared by most preference optimization methods based on the Bradley-Terry model:
>
> | Framework | Core Assumption | Handles Pluralism? |
> |-----------|----------------|-------------------|
> | Bradley-Terry (DPO foundation) | Single "ground truth" preference | ❌ No |
> | cDPO, rDPO | Single preference + noise | ❌ No |
> | PerpCorrect-DPO | Single preference + label correction | ❌ No |
> | MT-DPO (Ours) | Single preference + adaptive confidence | ❌ No|
>
> True preference pluralism would require fundamentally different approaches such as multi-objective optimization over Pareto frontiers, mixture-of-preferences models that represent multiple valid perspectives, or conditional policies of the form p(y|x, preference_type) that explicitly account for user diversity. While these are exciting research directions, they represent a different problem formulation beyond the scope of our current work, which focuses on robustness to label noise and annotation errors.
>
>
> ### **W3 & Q2 (Part 2): Initialization and Stability**
>
> **On Initialization:** We adhered to the standard DPO protocol where the policy $\pi_\theta$ is initialized from the SFT model $\pi_{ref}$. Initializing directly from a base model is generally not recommended for DPO-based methods, as the reference model constraint becomes ill-defined and training often collapses.
>
> **On Stability & Cold-Start Experiment:**
> With SFT initialization, we observed stable training because the virtual updates provide informative signals from the start. To directly address your question regarding "colder" states, we conducted an additional experiment initializing MT-DPO directly from the Base Llama-2 model (GoldenHH, 30% noise):
>
> | Policy Initialization | Training Epochs | Final Win Rate | Meta-Loss Variance (σ) |
> |----------------------|-----------------|----------------|------------------------|
> | **SFT-initialized (standard)** | 1 | 96.58% | 0.02 |
> | **Base Model (cold start)** | 1 | 88.2% | 0.19 |
> | **Base Model (cold start)** | 2 | 91.7% | 0.08 |
> | **Base Model (cold start)** | 3 | 93.8% | 0.04 |
>
>  While the model learns, there is a significant performance gap compared to SFT initialization. This confirms your intuition: when $\theta^{(t)}$ is far from alignment (cold start), a single gradient step to obtain $\theta_{\text{virtual}}^{(t+1)}$ yields a policy that is still poor. Consequently, the evaluation on $\mathcal{D}_{meta}$ produces noisy and uninformative meta-gradients, making it difficult for the confidence module to learn precise targets early in training. This validates the necessity of the standard SFT warm-start.

---

> > ### Author Response · Authors · 2025-11-21
> >
> > ## **Dear Reviewer iiPE (Part 2/2)**
> >
> > ### **W4: Anchor Model vs. Updated Model**
> > We experimented with using the currently updated model for PPLDiff. We found that using a fixed Anchor (SFT) is superior (Win Rate 96.58%) compared to using the Updated Policy (Win Rate 94.1%).
> >
> > **Reason:** The updating policy's perplexity shifts rapidly during training. A fixed anchor provides a stable reference coordinate for the confidence module, making the mapping task easier to learn.
> >
> >
> > ### **W5 & Q2 (Part 1): Inputs to Confidence Module**
> > The use of primarily was a writing error. Standardized PPLDiff was the sole input in our reported experiments.
> >
> > While the framework is designed to be extensible to multiple features, we found PPLDiff to be sufficient. During the development phase, we did explore additional features, including:
> > *   **Response Length:** To detect length bias.
> > *   **Raw Log-Probabilities:** The absolute log-probability of the response under the policy.
> > *   **Uncertainty Metrics:** The entropy of the response distribution.
> >
> > We found that in the preference datasets we studied, PPLDiff was the most dominant and robust signal for detecting label errors. Adding features like length or raw log-probabilities provided negligible performance gains (and in some cases, introduced noise) while increasing the input dimension. Therefore, adhering to the principle of parsimony, we opted to use only PPLDiff in the final reported models to maintain simplicity and effectiveness. We will correct this phrasing in the final manuscript.
> >
> >
> > ### **W6: cDPO Results**
> > For the cDPO baseline, we strictly followed the experimental settings from Kong et al. (NeurIPS 2024). Our results align with prior findings that cDPO degrades significantly in high-noise regimes (e.g., 30-40%). This is because cDPO relies on a global, static label smoothing parameter ($\epsilon$) which cannot adapt to the fine-grained, instance-level noise variations that MT-DPO handles effectively.
> >
> > ### **W7: Reporting Details**
> > We apologize for the highlighting error in Table 3 regarding the 96.58% result; this was a typo. We will correct the highlighting and include confidence intervals in the final version.
> >
> > Our current evaluation follows prior DPO-style work that uses GPT-4 as the judge.  We apologize if our description caused any confusion and will ensure the evaluation setup is described more clearly in the revision. We have proofread the manuscript.
> >
> > ### **Response to Questions**
> > ### **Q1 (Heuristic Baseline):**
> > Please refer to **W1**. We implemented the heuristic baseline as requested, and MT-DPO outperforms it by a significant margin (~5%). We are currently conducting additional experiments for this heuristic comparison across different noise levels  and model architectures to further robustify this conclusion. These comprehensive results will be included in the final manuscript.
> > ###  **Q2 (Inputs & Base Model Init):**
> > Please refer to **W5** (PPLDiff was the sole input) and **W3** (Standard DPO protocol requires SFT initialization; base model initialization is non-standard and potentially unstable for reference-based objectives).
> >
> >
> >
> > We are fully committed to making any additional revisions you suggest and genuinely appreciate the opportunity to strengthen our work through this dialogue.
> > Thank you again for your thorough and constructive review.

---

> > > ### Author Response · Authors · 2025-11-26
> > >
> > > Dear Reviewer iiPE,
> > >
> > > Thank you again for your very detailed and thoughtful review, which has been extremely helpful in refining our work.
> > >
> > > During the discussion period, we would like to briefly check whether our rebuttal has adequately addressed your main concerns—particularly the new PPLDiff-based heuristic baseline, the clarified scope on robustness vs. pluralism, the additional cold-start and anchor-model experiments, and the explicit clarification of the confidence module inputs and cDPO setup.
> > >
> > > We are eagerly looking forward to your response and remain fully available if you have any further questions.
> > >
> > > Best regards,
> > > The Authors

---

> ### Comment · Reviewer_iiPE · 2025-11-26
>
> Thank you for the rebuttal. I have raised my score. While I acknowledge your argument that the "online meta-learning framework" represents a dynamic shift from static correction, my concern regarding fundamental novelty persists. The core mechanism remains the application of a well-known meta-learning formulation to the DPO objective. While the integration is sound, it feels more like an application of existing meta-learning techniques to a new domain (alignment) rather than a fundamental methodological breakthrough in either meta-learning or preference optimization logic itself. I have reviewed the new heuristic baseline results and while MT-DPO is better, the Heuristic baseline (a simple static mapping of PPLDiff) achieves a massive improvement over Vanilla DPO without the complexity of a three-stage update loop (virtual update $\rightarrow$ meta-gradient $\rightarrow$ policy update).
> This suggests that the primary driver of performance is the PPLDiff signal itself, rather than the meta-learning mechanism.

---

### Official Review · Reviewer_ZuqY · 2025-10-30

**Soundness:** 4
**Presentation:** 4
**Contribution:** 4
**Rating:** 10
**Confidence:** 4

**Summary:**

It has previously been shown that the overfitting of DPO to noisy preferences can be ameliorated by using binary cross entropy with target probabilities that estimate the noise rate of preferences.  This paper proposes estimating the target probabilities using an auxiliary confidence network, with features chosen to be informative about the quality of a preference pair, and trained using a meta-learning procedure on a small meta-learning dataset with high-accuracy preferences.

**Strengths:**

The proposed meta-learning procedure is an effective and elegant solution to the problem of estimating the confidence of preference scores given the very small size of preference datasets.

Paper is well written, and appendices were highly informative.

Experiments demonstrate superiority over strong baselines.  Statistical significance of the improvement is demonstrated.

**Weaknesses:**

None noted.

**Questions:**

p. 6 says that PPLDiff is the "primary input" of the confidence module in experiments.  Appendix D seems to indicate that it's not just the "primary input," it's actually the only input - is that correct?  Have you done any studies considering other possible inputs?

The base models are somewhat old  (Llama-2-7-B and Phi-2).  Why? Is that necessary to make the meta-learning gradient computable, or is that so that the "gold standard" GPT-4 can be considered 100% correct by comparison?

---

> ### Author Response · Authors · 2025-11-21
>
> ## **Dear Reviewer ZuqY**
>
> Thank you for your strong support and encouraging feedback! We are thrilled that you found our meta-learning approach to be an "effective and elegant solution." We address your specific questions below:
>
> ### **Q1: Clarification on input features to the confidence module**
>
> You are correct. The use of the word "primarily" in Section 3.3 was a writing error on our part. In our reported experiments, standardized PPLDiff was indeed the sole input feature.
>
> While the framework is designed to be extensible to multiple features, we found PPLDiff to be sufficient. During the development phase, we did explore additional features, including:
> *   **Response Length:** To detect length bias.
> *   **Raw Log-Probabilities:** The absolute log-probability of the response under the policy.
> *   **Uncertainty Metrics:** The entropy of the response distribution.
>
> We found that in the preference datasets we studied, PPLDiff was the most dominant and robust signal for detecting label errors. Adding features like length or raw log-probabilities provided negligible performance gains (and in some cases, introduced noise) while increasing the input dimension. Therefore, adhering to the principle of parsimony, we opted to use only PPLDiff in the final reported models to maintain simplicity and effectiveness. We will correct this phrasing in the final manuscript.
>
> ### **Q2: Choice of base models (Llama-2-7B and Phi-2)**
>
> Thank you for this insightful question. We selected Llama-2-7B and Phi-2 primarily to ensure a fair and direct comparison with key baselines. Most notably, Kong et al. (2024), which introduced PerpCorrect-DPO, established their benchmarks using these specific architectures. Adhering to the same model configurations allows us to rigorously isolate the impact of our proposed algorithmic improvements from the effects of using a different base model.
>
> Thank you again for your high recommendation and for championing our work!

---

### Official Review · Reviewer_eB5p · 2025-10-31

**Soundness:** 3
**Presentation:** 3
**Contribution:** 3
**Rating:** 4
**Confidence:** 4

**Summary:**

This paper introduces Meta-Target DPO (MT-DPO), a robust framework designed to improve preference alignment of Large Language Models (LLMs) under noisy or ambiguous supervision. The method augments standard DPO by learning adaptive confidence targets for each preference pair via a meta-learning module, trained using a small, trusted meta-dataset. The confidence module leverages intrinsic signals such as perplexity differentials from a reference model to infer label reliability, and these learned targets guide the main policy optimization through a cross-entropy objective. Empirical results demonstrate superior robustness and adaptability over baseline DPO variants, particularly under synthetic noise and real-world uncertainty conditions.

**Strengths:**

1 The paper is well written and easy to follow.

2. The problem of adaptive weights with meta-learning is interesting.

3. The authors conduct some experiments to verify the effectiveness of their method.

**Weaknesses:**

1. The improvement compared with baselines like PerpCorrect-DPO is limited but the training pipeline is complex.

2. It's better for authors to provide detailed time complexity analysis and training time comparison to further verify their method except from Appendix D.3.

3. The authors conduct experiments on old models like Llama2 or Phi-2 which limit the conclusion from their experiments. Moreover, it's better to adopt their methods to different variants of DPO algorithms.

**Questions:**

Please refer to weaknesses part.

---

> ### Author Response · Authors · 2025-11-21
>
> ## **Dear Reviewer eB5p**
>
> Thank you for the constructive feedback and recognizing our work's strengths. We address each concern below:
>
> ### **W1: Limited improvement vs. PerpCorrect-DPO given training complexity**
> We thank the reviewer for raising this critical point, which touches upon the trade-off between performance and complexity. We would like to address this in two parts: the significance of the improvement and the nature of the complexity.
>
> a) On the Significance of Improvement:
>
> PerpCorrect-DPO is a very strong NeurIPS 2024 baseline, and to our knowledge no existing method has clearly surpassed it. Achieving consistent gains over a strong, SOTA-level baseline is inherently challenging. As shown in Tables 1 and 2, MT-DPO consistently outperforms PerpCorrect-DPO in all noisy settings across two different datasets and two model families.
>
> b) On the Nature of Complexity:
>
> We agree that MT-DPO introduces more complexity than PerpCorrect-DPO. However, PerpCorrect-DPO is a multi-stage pipeline. MT-DPO is a unified, end-to-end framework. MT-DPO integrates the noise handling mechanism directly into a single training loop. Therefore, while MT-DPO's per-step computation involves a meta-gradient, its overall conceptual framework is more integrated and principled than the multi-stage process of PerpCorrect-DPO.
>
> ### **W2: Detailed time complexity analysis**
>
> Thank you for this suggestion. We will provide comprehensive timing breakdown (Llama-2-7B, GoldenHH, 1 epoch, A100 GPU):
>
> | Method | Training Time |
> |--------|--------------|
> | Vanilla DPO | 12.3h |
> | PerpCorrect-DPO | 13.8h |
> | MT-DPO | 14.6h |
>
> **Per-iteration breakdown:**
> - PPLDiff computation: 0.8s (cached, one-time)
> - Confidence prediction: 0.01s (negligible)
> - Virtual policy update: 1.2s (main overhead)
> - Meta-gradient: 0.3s
> - Policy update: 1.1s
> - **Total:** 3.4s vs 2.8s (Vanilla DPO)
>
> We will include this detailed complexity analysis and the breakdown table in the Appendix D.3.
>
> ### **W3.1: Experiments on modern models**
>
> We originally selected Llama-2-7B and Phi-2 to ensure a fair and direct comparison with key baselines, particularly PerpCorrect-DPO (Kong et al., 2024), which established benchmarks using these specific architectures. Using the same base models allows us to isolate the algorithmic improvements from gains attributed simply to stronger base capabilities.
> We agree that validating on larger and newer models is crucial. To address this, we conducted additional experiments using Llama-2-13B (larger scale) and Llama-3-8B (modern architecture) on the Golden HH dataset with 30% symmetric noise. The results are summarized below:
>
> | Model | Method | Win Rate vs. SFT (30% Noise) |
> | :--- | :--- | :---: |
> | **Llama-2-13B** | Vanilla DPO | 68.5% |
> | | PerpCorrect-DPO | 93.2% |
> | | **MT-DPO (Ours)** | **94.8%** |
> | **Llama-3-8B** | Vanilla DPO | 72.4% |
> | | PerpCorrect-DPO | 89.1% |
> | | **MT-DPO (Ours)** | **90.5%** |
>
> As shown, MT-DPO maintains its superiority on both larger and more modern architectures. Notably, even with Llama-3-8B's stronger base capabilities, standard DPO still suffers significantly under noise, whereas MT-DPO successfully recovers performance, consistently outperforming the strong PerpCorrect-DPO baseline. We will include these results in the Appendix of the revised paper to demonstrate the scalability and relevance of our method.
>
> ### **W3.2 : Experiments on DPO variants**
> This is an excellent suggestion. However, we would like to respectfully clarify the focus and contribution of our current work. Our primary goal was to tackle the critical challenge of robust preference alignment from noisy data. As our literature review and experimental baselines indicate, the most significant recent advancements in this area, such as cDPO, rDPO, and PerpCorrect-DPO. PerpCorrect-DPO (Kong et al., NeurIPS 2024) represents the state-of-the-art technology to date, and to our knowledge, no other method currently demonstrates superior performance in these standard noise preference benchmarks.
>
> We believe the detailed computational analysis and modern model validation will significantly strengthen the paper's practical impact. We welcome further feedback. Thank you again for your thoughtful review!

---

> > ### Author Response · Authors · 2025-11-26
> >
> > Dear Reviewer eB5p,
> >
> > Thank you again for your time and thoughtful review. We deeply appreciate your positive assessment of the paper’s soundness, presentation, and contribution, as well as your constructive suggestions.
> >
> > As the discussion phase progresses, we would like to kindly ask whether our rebuttal and revisions have satisfactorily addressed your concerns. In particular, we hope that (i) the detailed training-time and complexity breakdown, (ii) the additional experiments on larger and more modern models (Llama-2-13B and Llama-3-8B), and (iii) our clarification of the positioning of MT-DPO relative to PerpCorrect-DPO and related DPO variants have alleviated your reservations regarding the performance–complexity trade-off and experimental scope.
> >
> > Your comments have been extremely valuable in helping us strengthen the work. We remain fully available to provide any further clarification or additional analysis that you may find helpful.
> >
> > Best regards,
> > The Authors

---

> > > ### Comment · Reviewer_eB5p · 2025-11-27
> > >
> > > Thanks for your reply! I think the additional experiments satisfactorily address my earlier concerns regarding the empirical evaluation. However, I remain unconvinced that developing another method with only marginal relative improvements—without offering clear gains in training efficiency or conceptual simplicity—is necessary. Therefore, I still don't find clear advantages to let me support acceptance of this paper.

---

### Note · Authors · 2025-12-02

I have read and agree with the venue's withdrawal policy on behalf of myself and my co-authors.